# Small Models, Smarter Learning: The Power of Joint Task Training

## Abstract

The ability of a model to learn a task depends critically on both task difficulty and model size. We study this relationship for compositional operations, focusing on nested ListOps and extending beyond arithmetic to permutation groups, with the goal of determining how task difficulty sets the minimum parameter requirements for small transformer models. We vary task difficulty by introducing new operations or combinations of operations into the training data. We find that while operations such as modular addition or permutation group products are difficult in isolation, joint training with other operations, including product, maximum, or auxiliary sub-block operations, reduces the parameter requirements by factors of 2 to 7. Analysis of learned embeddings using PCA reveals that when joint training helps it is usually accompanied by an increase in highly regular structures in the embedding of inputs. These results suggest that joint training leads to qualitatively different learning trajectories than learning operations in isolation, with shared number representations supporting difficult tasks such as addition. Our findings further demonstrate the importance of training curriculum on the emergence of abilities in language models.

## 1 Introduction

Scaling laws for language models Kaplan et al. (2020) characterize how performance scales with model size (number of parameters) Hoffmann et al. (2022), compute (FLOPs) Muennighoff et al. (2024), dataset size (number of examples) Hestness et al. (2017), and information capacity (in bits) Allen-Zhu & Li (2024). They have been used as a guide for model design, predicting the best loss achievable for a given setting. Moreover, the study of emergent abilities in LLM showed Wei et al. (2022) that various capabilities emerge at different model sizes. Both of these directions suggest an intimate and universal relationship between model size and the emergence of abilities. However, these views miss important details about the role of the training curriculum.

In general, language models performs poorly on symbolic mathematical tasks Frieder et al. (2024); Dziri et al. (2024); Dave et al. (2024) such as the ListOps dataset Nangia & Bowman (2018) used in this study, which consists of nested math operations. Models often struggle with generalization, tending to memorize tasks rather than simulate the underlying algorithms. While mathematical tasks prove challenging for large language models to learn, they provide a controllable playground to test how models learn different tasks and to evaluate their accuracy quantitatively. Furthermore, they allows us to tune the task difficulty by combining different operations.

Models may learn different algorithms for solving a task. Yet, it is unclear if, similar to human, under what conditions models may brute-force memorize a task, vs learn generalizable solutions. In this work, we show strong evidence that the learning curriculum can dramatically shift the onset of emergence of abilities in small untrained models. Models of the same size may follow different learning paths depending on the training setup, suggesting that the tasks themselves may change the the way learning dynamics finds a solution.

It is worth noting that our observations point to more than just task complexity being the driving force. Complexity of a task may be defined using Kolmogorov complexity (KC) Kolmogorov (1965); Li et al. (2008), defined as the length of the shortest code producing a desired output. What we show is that models may require vastly different minimum sizes to learn the same task, depending on the training curriculum. Additionally, we show evidence of a potential change in the internal algorithm

learned by the models using different curricula. Such a distinct algorithm could represent learning different algorithms to solve the same problem, with each algorithm having a different resource requirement, e.g. brute-force versus an efficient code. Hence we argue the model size depends more on the *algorithmic complexity*, or description length of a specific algorithm, rather than the KC of the task. This also raises an important question: what training regiments are most effective at inducing the learning of more efficient algorithms?

Using the ListOps dataset as our experimental framework, we investigate how small transformer models learn nested mathematical operations—specifically maximum (MAX), minimum (MIN), median (MED), and sum modulo $n$ (ADD, aka SUM)[1], extended with product modulo $n$ (PROD) and alternating sum and subtraction modulo $n$ (NADD:= $\sum_i (-1)^i x_i$ ). We further introduce permutation groups as an additional class of problems, where group products can be decomposed into products of subgroups, such as sub-block operations. We train the models on a single or a mixture of these operations. We make the following observations:

1. **Easy tasks:** MAX and MIN are easiest, followed by PROD (non-prime $n$) and MED.

2. **Hard tasks:** ADD and NADD (mod any $n$), and PROD mod primes are considerably harder, requiring and order of magnitude larger models.

3. **Joint training paradox:** surprisingly, mixing some easy tasks with hard tasks leads to smaller models mastering the hard tasks. In contrast, mixing hard tasks did not seem to help in our experiments.

4. **Beyond arithmetic:** We show the same effect also in block-diagonal permutation matrices, demonstrating that the effect of joint training is not restricted to the arithmetic setting.

We hypothesize that compared to pure ListOps tasks, the mixed training makes it easier for the models to discover number properties of the symbols. This seems to lead the mixed models learning a different algorithm than the pure model. For instance, we observe a similar difficulty in learning pure ADD for models learning a randomized ADD dataset, where we create a randomized sum table with the right-hand side of $A + B = C$ being shuffled (Appendix D). In the randomized ADD task, any number relation between the symbols has been erased, and the easiest way to learn it should be to memorize the sum table, rather than overfit the random patterns in the sum table.

These results provide compelling evidence that joint training guides models towards finding alternative, more efficient solutions. They suggest there is more nuance to the scaling laws and they can be significantly affected by the training curriculum and strategies.

## 2 RELATED WORK

Multi-task learning helps models generalize by sharing useful features across related tasks. For compositional problems, joint training allows models to develop basic building blocks, leading to sudden jumps in performance, as shown by Abedsoltan et al. (2025). Similar results by Lee et al. (2024); Wang & Lu (2023) show that training transformers on several arithmetic operations improves each task by encouraging shared number representations and multi-step reasoning. Curriculum learning—training on easier tasks before harder ones—further strengthens mathematical reasoning Yin et al. (2024); Kim & Lee (2024); Jia et al. (2025).Work on grokking Power et al. (2022); Liu et al. (2022) and mechanistic interpretability Nanda et al. (2023) shows that models solving modular arithmetic form highly structured internal embeddings. While prior work has explored multi-task and curriculum learning, these studies do not examine how model size requirements change under different curricula, nor do they identify which specific task pairings succeed or fail. Although we use a scratch-pad setup similar to grokking studies, our focus is on training small models on combinations of arithmetic tasks. Building on this, we measure how task composition affects the minimum model size required for learning and identify which task combinations help or fail to help models learn hard tasks.

---

[1]we will use SUM and ADD interchangeably

## 3 METHODOLOGY

We train small-scale transformer models and analyze both their performance and internal representations. This section outlines our methodology, detailing the dataset, model architecture, and evaluation protocols.

**Choice of task**   ListOps consists of nested mathematical equations involving operations. Our setup uses MAX, MIN, MED (median), ADD (sum modulo $n$), PROD (product modulo $n$) and NADD (alternating sum and subtraction modulo $n$) applied to numbers (0-$n-1$). We conduct experiments using $n = 10$ to $n = 226$, (Appendix F.3.1). Choosing ListOps was motivated by several key factors that align with our research objectives:

1. **Procedural Generation:** ListOps is a synthetically generated dataset, allowing us to create a large volume of examples with controlled difficulty.
2. **Exact Evaluation:** The mathematical nature of ListOps operations ensures unambiguous, exact evaluation of model outputs.
3. **Adjustable Task Complexity:** ListOps offers a framework to modulate problem complexity by combining different operations and adjusting nesting depths.
4. **Inter-task Relationships:** The dataset's multiple operations provide an opportunity to explore task synergies and interference.

**Dataset Description**   We use a simplified functional notation of the form $f(x, y, \ldots)$. For example, `max_10(3,min_10(7,4,9))=4<eos>`, where `<eos>` denotes the end token. The subscript in the function name indicates the modulo operation; for instance, `max_10` represents the `max` function modulo 10.

**Tokenization and CoT:**   We employ a character-based tokenization strategy for processing ListOps expressions. (Appendix F.3.2). We find that directly solving nested ListOps in one step can be quite challenging for transformer model (Fig. 23) Even with a **maximum of three nesting levels with three operands (inputs)** we find that GPT models with over 10 million parameters still fail to learn the task. To enhance model performance, particularly on more complex operations like sum modulo 10, we introduced a chain of thought (CoT) approach in our training data (Appendix F.3.3): `add_10(1,2,add_10(3,4))>add_10(1,2,7)>0=0<eos>`, where the '>' token means one step of CoT, wherein we solve the right-most, inner-most parenthesis. This is similar to the *scratchpad* used in Lee et al. (2024), among others. The tasks become increasingly challenging to learn with increasing nesting, number of operands, and length in a predictable manner (Appendix Fig. 44).

**Train Test Split:**   The experiments were conducted on data generated with a maximum nesting depth of 2 and up to 3 operands per operation. The equations are constructed from the combination of digit sequences forming duplets and triplets (e.g., "1,0", "0,0,1"). The CoT could result in many patterns appearing in one equation. To ensure that all the patterns in the test set are not also present in the training data, we select 100 out of the possible 1000 triplets of numbers, e.g. 749, to make an exclusion set. The training data is chosen from equations which never encounter the triplets in the exclusion set, and the test set is chosen such that each equation contains at least one excluded triplet.

**Permutation groups**   Modulo addition forms a cyclic group, which is itself a subgroup of the permutation group. The matrix representation of permutations provides a natural framework for defining subtasks through operations on blocks or submatrices. By Cayley's theorem, every finite group can be represented as a subgroup of permutations. Extending this idea, finite groups can be directly incorporated into ListOps tasks, since they are closed under group operations. To explore this, we introduce a new task of learning the product table of permutation matrices. Specifically, we construct $6 \times 6$ block-diagonal permutation matrices composed of two $3 \times 3$ blocks and define three operations: `OP`, which acts on the full $6 \times 6$ matrix; `OP_TOP`, which acts only on the top block; and `OP_BOTTOM`, which acts only on the bottom block (Appendix Fig. 12, 13, 15, 14). The block-diagonal group contains 36 elements (Appendix Fig. 11), and we implement the task in the ListOps formalism. For example: `OP(1,2,OP_TOP(2,3)) >`, where each number corresponds to an element of the group which is a matrix.

**Performance Evaluation:** We randomly select 1000 equations from the held-out test set. The model is prompted to generate solutions character-by-character, starting from the equation prompt (e.g., `add_10(3,min_10(7,4,9))>`). We use the output to produce two metrics: **Loss:** using cross-entropy computed for every character of the output. **Accuracy:** based on the number of correct answers evaluated using *only the final answer*, which we define as the first character after the first '=' symbol.

**Model Architecture** For our experiments, we employed a series of tiny GPT models, inspired by the nanoGPT architecture (Karpathy, 2022). Unlike the standard sequential transformer, we implemented a recurrent variant in which a single transformer block was iteratively applied by feeding back its output as the next input (see the architecture Appendix Fig. 22). This recurrent design improved learning efficiency and yielded more structured embedding patterns. Each model in our study uses a single attention head. We set the feedforward hidden dimension to four times the embedding dimension, a common practice in transformer architectures, providing sufficient complexity in the feedforward networks while keeping the model size constrained. By varying the embedding size we create a range of models with different parameter counts.

## 4 RESULTS

To understand how language models acquire mathematical abilities, we focus on accuracy as the primary metric for observing the emergence of learning, following the approach of the emergent abilities literature as in Wei et al. (2022). While some subsequent studies have questioned the concept of emergence by examining other metrics (Schaeffer et al., 2024), we argue that this critique overlooks similar patterns observed in physics: in a phase transition, only some quantities may exhibit discontinuous changes. A metric that can capture the emergence of the new phase is called the "order parameter". Hence, we choose to use accuracy as our order parameter.

We run experiments with embedding dimension $n_{embed}$ between 8–362. Similar to the emergence literature, we observe the total number of parameters to be the strongest indicator of accuracy, rather than $n_{embed}$ or depth. We primarily present results for the modulo 20 experiments, but we observed similar patterns for other moduli (Appendix F.5 - mod 10 results, F.4 - mod 26 results). Furthermore, we show that model performance scales with the modulus, as higher moduli require larger models and more data to learn effectively.

Importantly, joint training on compositional tasks substantially improves performance and reduces the parameter requirements for learning individual operations. Training on multiple arithmetic tasks helps models learn the basic building blocks needed for solving more complex problems. This shared representation not only accelerates learning but also produces abrupt transitions in performance once the necessary constituent skills are mastered Lubana et al. (2024); Okawa et al. (2023). Such effects, consistent with previous findings on compositional generalization Lee et al. (2024), demonstrate the utility of joint training for scaling model competence on algorithmic tasks.

**Transition to Learning.** Figure 1 shows the learning transitions for three single operations and three joint operations. We find that joint training enables models to solve more difficult tasks with fewer parameters and less data. For example, ADD and NADD are the most demanding tasks when trained in isolation, yet when combined with PROD, models that are 2.5 times smaller succeed. Similarly, while PROD on prime moduli is challenging on its own, pairing it with MAX reduces the required model size by a factor of 2.6.

These results challenge the common assumption that harder tasks always demand larger models. Instead, something emerges during joint training that lowers the effective complexity of learning. Similar effects have been reported in prior work on language models, where training on diverse tasks improves generalization Lee et al. (2024). Our results suggest that in ListOps, mixing tasks likewise produces a dramatic reduction in the learning threshold, raising the question of what underlying mechanisms drive this synergy.

**Embedding Layer.** To understand the difference between the hierarchy of transition points observed, we first examine the embedding layer of the models which learned the tasks (acc. > 90%) (see Appendices F.4 and F.5 for more). While the $n_{vocab} \times d_{embed}$ embedding layers of models of

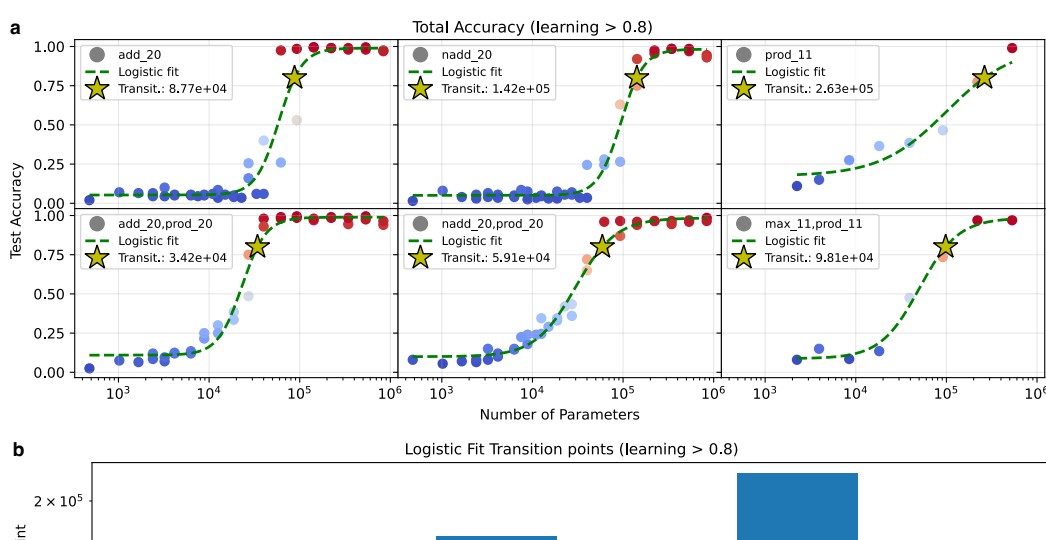

Figure 1: **Joint training. (a)** Each panel shows the same group of small transformer models trained on different operations, either in isolation or in combination. The first column compares ADD with ADD+PROD, the second column compares NADD with NADD+PROD, and the third column compares PROD with MAX+PROD. The top row shows models trained on individual operations, while the bottom row shows the corresponding joint-training results. Red dots indicate models achieving more than 50% accuracy, and blue dots indicate models below 50%. The dashed green line is a logistic fit, and the yellow star marks the transition point at 80%. The x-axis denotes model size (number of parameters), and the plots are ordered by increasing transition point. Training on individual operations is challenging, but joint training reduces the required model size to learn the tasks. **(b)** Bar plots of the model sizes at transition points, with training on individual operations shown in blue and joint training in red.

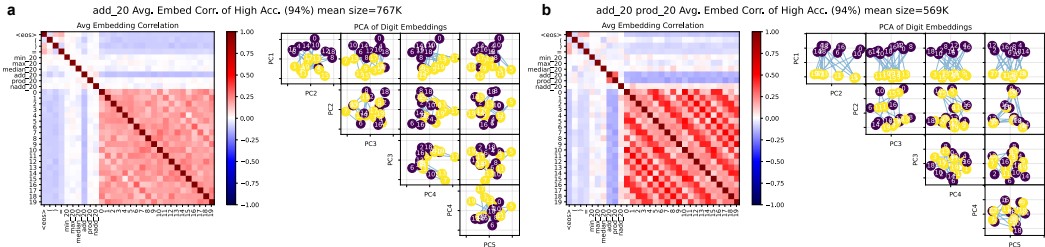

Figure 2: **PCA of embeddings**: We selected all models that achieved over 90% test accuracy. For each operation and operation combination, we show the average correlation matrix and the top principal components (PCs) of the cosine similarity of the embeddings. The PCs are colored by number parity (odd numbers in yellow, even numbers in purple). **(a)** ADD: the model does not capture a regular pattern; however, PC1 and PC2 separate odd and even numbers, indicating that parity is an important feature for the sum in MOD 20. (See more results for other MOD in the Appendix I) **(b)** Joint training on ADD and PROD: the PCs reveal a clear structure, separating odd and even numbers, grouping similar numbers together, and even forming grid-like patterns.

different sizes cannot be directly compared, we can compare them using $n_{vocab}$ dimensional principal component analysis (PCA). Moreover, while the PCs of individual models may be noisy, we can find aggregate PCs by first computing the $n_{vocab} \times n_{vocab}$ correlation matrices of a group of models and averaging them. This approach not only reduces the noise but also allows us to combine models of different sizes.

An interesting picture emerges when we average models which learned each individual operation. Figure 2 shows the correlation matrix and top PCs for the embeddings of models trained on the groups we identified in the transition plot: **a**) ADD, **b**) ADD + PROD. We note that the principal components for the ADD operation exhibit a noisy pattern; however, in PC1 and PC2, the model nearly perfectly separates the parity of the numbers (Fig. 2a), indicating that parity is a key feature for learning the modulo addition operation. When ADD and PROD are combined, all principal components display an organized pattern. PC1 clearly separates number parity and also groups numbers modulo 4, for example, 0, 4, 8, 12, 16 and 1, 5, 9, 13, 17 (Appendix 8). The remaining PCs exhibit a lattice-like structure, grouping different pairs of the modulo-4 classes.

This example illustrates that joint learning of multiple operations can be mutually beneficial, enabling the model to develop more intuitive embeddings that support both tasks—particularly those that are difficult to learn in isolation. Here, the PROD operation clearly separates number parity (Appendix 9), providing a bias that assists the ADD operation, for which parity appears to be a key feature. This guidance from PROD allows ADD to be learned effectively with smaller models.

Given that much smaller models learn ADD when jointly trained with PROD than when trained on ADD alone, we hypothesize that the two groups of models employ different algorithms for ADD—or at least follow distinct learning trajectories. Mixed training appears to guide the model toward an understanding of number properties, enabling it to employ efficient and compact algorithms for ADD. In contrast, models trained solely on ADD tend to represent symbols as numbers. Moreover, modulo $n$ ADD tables exhibit a uniform distribution of elements—a notoriously difficult pattern to learn. Introducing another operation disrupts this uniformity, which may facilitate learning the task. However, we hypothesize that merely breaking the symmetry is insufficient, and that deeper shared features between the two operations make joint training particularly powerful. One way to test the hypothesis of Pure ADD finding symbolic solutions is to design a similar problem where combination of symbols $(A, B)$ turns into $C$, analogous to a shuffled sum table.

**Shuffled ADD.** We conducted experiments using a shuffled, symmetric ADD table ($A+B = B+A$) in modulo 26, comparing a model trained on ADD alone with one trained on MAX, MED, and Shuffled ADD. The shuffled addition proved more difficult to learn than all other mathematical operations except the original ADD, which remained slightly harder even than the shuffled version (Fig. 3). Similarly, MAX+MED+Shuffled ADD was more challenging than all tasks except pure ADD, suggesting that inherent number properties facilitate learning in mixed training.

Analysis of the PCA embeddings revealed that most of the features observed with regular numbers were absent, aside from partial ordering, as MAX and MED continued to rely on standard number ordering (Appendix Fig. 21). Mixed training provided little advantage for Shuffled ADD: the task reached only 60% accuracy and required twice as many parameters compared to a model trained solely on Shuffled ADD, highlighting the importance of internal number representations for solving addition. PCA embeddings for Shuffled ADD likewise showed no structure comparable to that observed in the mixed model with regular numbers. Coloring the numbers by parity further confirmed this effect: unlike in the regular mod-26 experiments, no clear parity separation emerged (Appendix Fig. 21), as expected.

**Learning dynamics Pure ADD vs MAX+MED+ADD.** Another piece of evidence for the difference between ADD and the rest comes from the training dynamics. Figure 4 shows the loss curves and the evolution of test accuracy in a pure ADD model vs MAX+MED+ADD, for models with $n_{embed} = 96$ and 3 layers, which is slightly above the pure ADD learning transition. The mixed model was evaluated separately for each of the MAX, MED, and ADD test sets to see when each ability emerged. We observed a subtle but important sequence in learning: MAX was learned fastest, followed closely by MED, and then ADD. Crucially, the gaps between the learning of these operations were minimal, with the model beginning to grasp ADD while still perfecting MED. The accuracy curves also show that joint training accelerates the learning of the ADD operation, as evidenced by

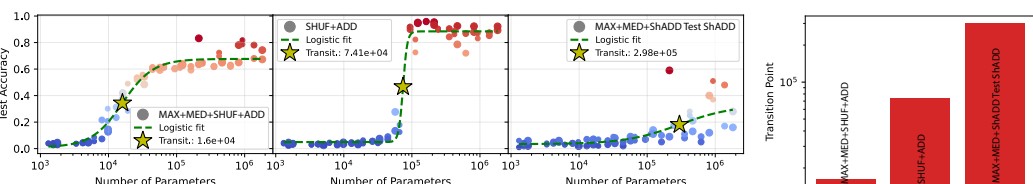

Figure 3: **Shuffled symmetric sum table, Mod 26, 40K steps:** We find that this shuffled version of sum is again more difficult to learn than any of the math operations except for the original Pure ADD, which remains slightly more difficult than even the shuffled version. We also observe that MAX+MED+Shuffled ADD never reaches more than 80% accuracy. The third scatter plot from the left shows the accuracy of the MAX+MED+Shuffled ADD model on the Shuffled ADD test set. We see that the accuracy is very low ($\sim 20\%$ top), showing that the mixed model never learned the shuffled ADD. This may suggest that MAX+MED revealed number properties, but Shuffled ADD was incompatible with those properties, leading to a model that overall cannot solve the two problems (MAX+MED and Shuffled ADD) simultaneously.

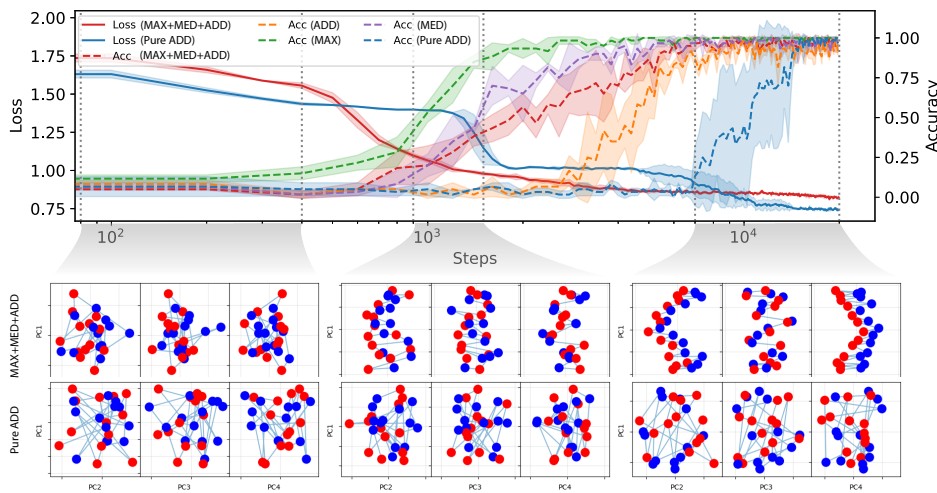

Figure 4: **Evolution of training loss, accuracy, and PCs of the embedding layer, Mod 26:** The top shows the evolution of training loss (solid lines) and test accuracy (dashed lines) for models with an embedding dimension of 96 and 3 layers, trained either on pure ADD (blue) or on mixed MAX+MED+ADD (red). Models trained on MAX+MED+ADD were evaluated separately on pure MAX, MED, and ADD subsets; the corresponding accuracy curves are shown in green, purple, and orange, respectively. Curves are the mean on 3 runs, with shaded $\pm\sigma$. All models were trained for 20000 iterations. The figures beneath the main plot display PCA embeddings revealing that models trained on MAX+MED+ADD data progressively develop a structured representation of numerical concepts, accompanied by a steady decrease in loss. The PCAs also show a prominent parity separation emerging in PC2 and PC4. Parity is colored by red and blue. In contrast, models trained solely on ADD exhibit no clear structure in the embedding space and show long plateaus in the loss curve.

the much earlier rise of the orange dashed line (joint training) compared to the blue dashed line (pure ADD training), occurring at almost half the number of steps of pure ADD.

**Joint Training May Shrink the Search Space.** The model trained on mixed data can take a very different route and converge to a different solution. We see that training on MAX, MIN, or MED all lead to embeddings which exhibit number properties. Because of the CoT steps, the model's loss decreases when it learns any of the operations involved in the mix. For instance, since learning

MAX is much easier than ADD, gradient descent may first learn to solve MAX, which is strongly associated with learning a representation for the digits with the correct ordering. This restriction of the embedding layer can make it significantly easier for the model to learn other number properties and, possibly, find a number-based algorithm for ADD, exploiting the properties of numbers instead of memorizing symbolic patterns.

## 4.1 TESTING THE EMBEDDING RESTRICTION HYPOTHESIS.

We test the hypothesis that restricting the embedding to what is learned by operations like MAX and MED could lead to smaller models learning ADD. To further explore this hypothesis, we designed a transfer learning experiment: 1) We first trained a model ($n_{embed} = 48$ and three layers, which on pure ADD was unable to learn) on MAX and MED operations until proficiency. 2) We then gradually introduced ADD operations, increasing their proportion in the training data from 0% to 100% over 1000 steps, while simultaneously phasing out MAX and MED.

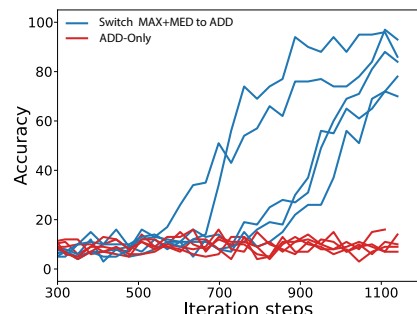

Figure 5: **Learning ADD by fine-tuning MAX+MED:** We train model much smaller than the ADD-Only learning transition (48 embedding, 2 layers - traditional Transformer architecture). By switching the training data slowly from MAX+MED to pure ADD (never showing expression mixing all three) the model is able to learn ADD (blue) in this much lower parameter regime. In comparison, the pure ADD models (red) did not learn at this size.

Key findings from this experiment include:

- The model began learning ADD immediately upon its introduction, despite never seeing mixed expressions (e.g., ADD with MAX or MED).

- Interestingly, the model started to forget MAX and MED once these operations were no longer present in the training data.

- Crucially, we verified that the model retained its number-like embedding structure even after MAX and MED were completely phased out.

Perhaps most strikingly, we found that even much smaller models (embedding size 24) could learn ADD perfectly using this hybrid approach, mirroring the efficiency of MAX+MED+ADD models. This resulted in a model capable of performing ADD operations that was 7x smaller than the PURE ADD model, while relying on a more sophisticated, number-like internal representation.

## 4.2 MODEL SCALING WITH MODULO N.

We also analyzed how the model size required to learn a given operation—or combination of operations—depends not only on the nature of the task but also on the modulo. In other words, we asked how the model scales when using different numerical systems. Models were trained on tasks including `PROD_n`, `ADD_n+MAX_n+PROD_n`, `MAX_n+PROD_n`, `ADD_n+PROD_n`, and `ADD_n`, and the transition point (defined as the smallest model reaching 80% of the learning curve) was measured and plotted against the modulo n. To ensure sufficient training data, the dataset size was scaled linearly with n (Fig. 6). We find that learning the PROD operation modulo prime numbers is particularly challenging, likely due to the uniform distribution of values in the product table. We also show that the model does not converge to a well-structured representation in the PROD–mod–prime case (Appendix Fig. 10), further demonstrating the hardness of this task. Interestingly, joint training with MAX facilitates learning of PROD. Similarly, ADD consistently requires larger models when trained alone compared to ADD+PROD in the non-prime case across all input range (modulo). For ADD, combining it with MAX and MED operations also accelerates learning.

## 4.3 PERMUTATION GROUPS

We generalize the ListOps framework to finite permutation groups by leveraging their closure under group operations. To demonstrate this, we construct a task where models learn the product table of $6 \times 6$ block-diagonal permutation matrices, composed of two $3 \times 3$ blocks. We define three operations: `OP`, acting on the full matrix; `P_TOP`, acting only on the top block; and `OP_BOTTOM`,

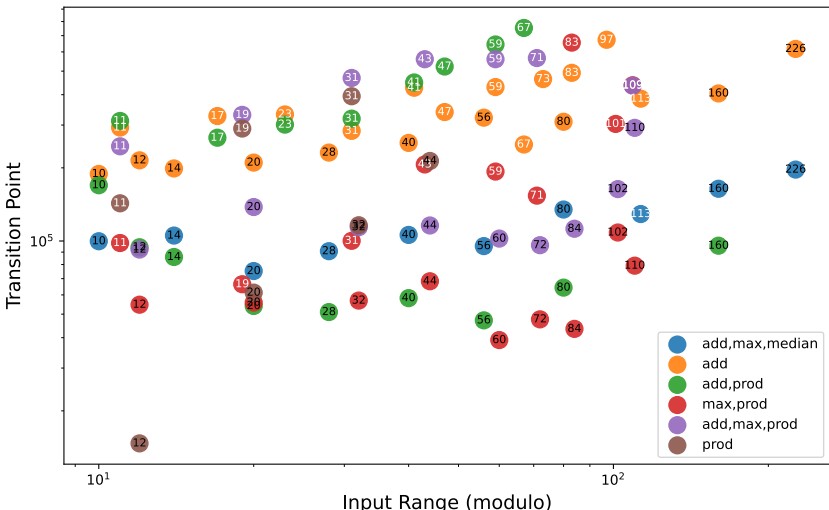

Figure 6: **Scaling of transition point vs input range (modulo)** Learning PROD modulo prime numbers (white font) is challenging due to its uniform distribution, but joint training with MAX facilitates learning. ADD also benefits from combinations with PROD or MAX+MEDIUM, requiring smaller models than when trained alone. Of note, is that mixing hard tasks such as ADD and PROD at primes does not lead to easier learning. Also, ADD+MAX+PROD at primes is often harder than PROD or ADD.

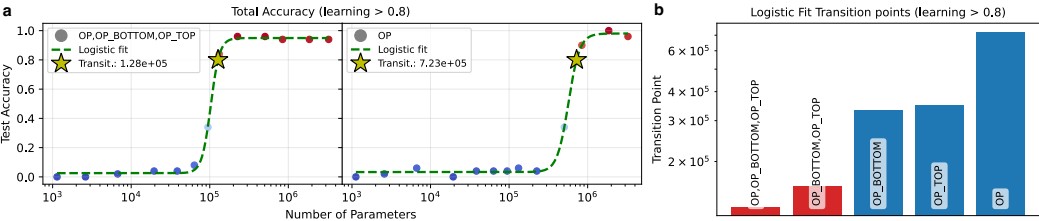

Figure 7: **Permutation - block diagonal with two 3x3 block matrix** Learning 6x6 permutation subgroup with two 3x3 diagonal blocks becomes significantly easier when jointly trained on operations on the top and bottom blocks, leading to 5x smaller models mastering the 6x6 group operations.

acting only on the bottom block (Appendix Fig. 12). The resulting block-diagonal group contains 36 elements, and the task is expressed in ListOps notation, for example, `OP(1,2,OP_TOP(2,3))`, where each number indexes a group element represented as a matrix.

We find that learning the three operations jointly, `OP`, `OP_TOP`, and `OP_BOTTOM`, facilitates learning of `OP`, enabling a 6x reduction in model size (Fig. 7). This provides another example that the benefits of joint training extend beyond arithmetic operations.

## 5 DISCUSSION AND LIMITATIONS

While the benefits of mixed task training have been known empirically, the exact mathematical mechanism behind it not yet known. Our work elucidates some aspects of this effect. Our results suggest benefiting from joint training often coincides with better representation learning for the tokens. This may be due to secondary easier tasks restricting the search space for correlated embeddings of related tokens, e.g. PROD and MAX revealing properties of numbers, leading to better pattern matching for ADD.

Such improved embeddings had also been observed in the "grokking" literature Power et al. (2022); Liu et al. (2022) revealing that successful generalization coincides with emergence of highly organized internal representations. We suspect that better starting points may lead to better learning. If models

begin with structured embeddings or learn them through carefully chosen auxiliary tasks, they may solve problems more efficiently. This suggests that the path to learning matters as much as the final destination: models that develop organized internal representations early in training may require fewer resources to master difficult tasks.

Our results on permutation groups suggest this effect is not restricted to arithmetic tasks. Indeed, using Cayley's theorem, which states every finite group is a permutation subgroup, we can argue that understanding the effect of mixing tasks in the context of permutations could encompass a large class of verifiable tasks on finite sets. We believe that a systematic study of permutation groups, could reveal more about the conditions required for joint training to be beneficial. Additionally, many operations, such as MED, and the OP_TOP we defined n permutations, are not group operations and require more investigation.

While the paper focuses on arithmetic operations, we hypothesize that joint training could be beneficial in more realistic scenarios, such as combining arithmetic with language tasks. Prior research has shown that training arithmetic operations together with text improves model accuracy, and that training on cellular automata patterns can help models perform downstream tasks such as chess move prediction (Yin & Yin (2024); Lee et al. (2024); Zhang et al. (2024). These examples suggest that models can develop underlying, fundamental representations that can be leveraged to improve performance on downstream tasks.

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

## A  RELATED WORKS

**Scaling laws** for language models characterize how validation loss scales with model size (number of parameters) Hoffmann et al. (2022), compute (FLOPs) Muennighoff et al. (2024), dataset size (number of examples) Hestness et al. (2017), and information capacity (in bits) Allen-Zhu & Li (2024), offering a foundation for model design. However, this perspective overlooks critical aspects of learning. Our results show that the composition of the training data—specifically, the mix of tasks—can significantly alter what and how a model learns. Even models with identical sizes can follow different learning trajectories depending on the training setup, suggesting that task structure shapes the internal algorithms that emerge during training.

**Joint training** Joint training on compositional tasks allows models to acquire the foundational primitives necessary for solving complex operations, often leading to abrupt performance improvements once all constituent skills are learned Lubana et al. (2024); Okawa et al. (2023). Similarly, training on multiple arithmetic tasks has been shown to improve accuracy on individual operations, highlighting the benefits of shared representations for compositional generalization Lee et al. (2024).

**Number representation** Neural networks trained on modular addition tasks develop structured embedding spaces that reflect underlying arithmetic operations. Mechanistic analyses have shown that small models can exhibit ordered number patterns and implement distinct algorithmic strategies, depending on initialization and hyperparameters Zhong et al. (2023). Some models converge to known solutions such as the Clock algorithm, while others discover novel procedures like the Pizza algorithm, illustrating the algorithmic diversity that can emerge from fixed training data. Periodic structures in the embeddings can be characterized via Fourier analysis, offering additional interpretability Nanda et al. (2023). These behaviors have also been linked to grokking dynamics, where models abruptly generalize after extended training, accompanied by the emergence of structured embedding patterns Power et al. (2022); Liu et al. (2022).

In general language models performs poorly on symbolic mathematical tasks Frieder et al. (2024); Dziri et al. (2024); Dave et al. (2024) such as the ListOps dataset Nangia & Bowman (2018) used in this study. Models often struggle with generalization, tending to memorize tasks rather than simulate the underlying algorithms. While mathematical tasks prove challenging for large language models to learn, they provide a controllable playground to test how models learn different tasks and to evaluate their accuracy quantitatively. Furthermore it allows us to tune the task difficulty by combining different operations.

Here, we investigate how language models acquire arithmetic skills by training small models on the ListOps dataset, which enables explicit evaluation through structured mathematical expressions. Adopting a bottom-up approach, we find that models learn to solve these tasks once the number of trainable parameters surpasses a critical threshold. This threshold shifts with the difficulty of the target operation (MAX = MIN < MED < SUM), indicating a dependency on task difficulty. Surprisingly, joint training on multiple operations often facilitates learning, outperforming models trained on individual operations (e.g. MAX+MED+SUM < SUM). This suggests that task diversity can ease optimization by promoting shared representations.

# SUPPLEMENTAL MATERIAL

# B ADDITIONAL PLOTS

## B.1 MODULO 20

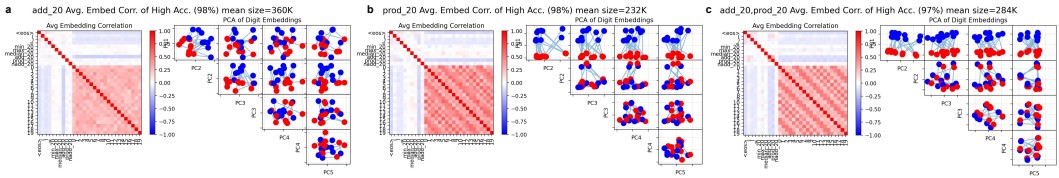

Figure 8: **ADD+PROD embeddings and PCs. PCs colored based modulo 4.**

Figure 9: **ADD, PROD, ADD+PROD embeddings and PCs.**

## B.2 MODULO PRIME - PROD

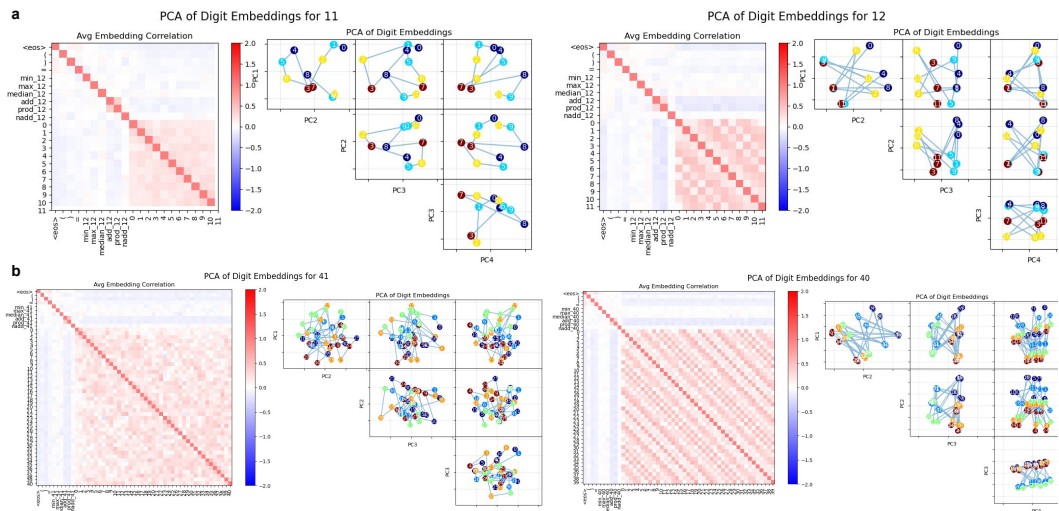

Figure 10: **PCs PROD modulo prime vs not prim.**

## C PERMUTATION GROUPS.

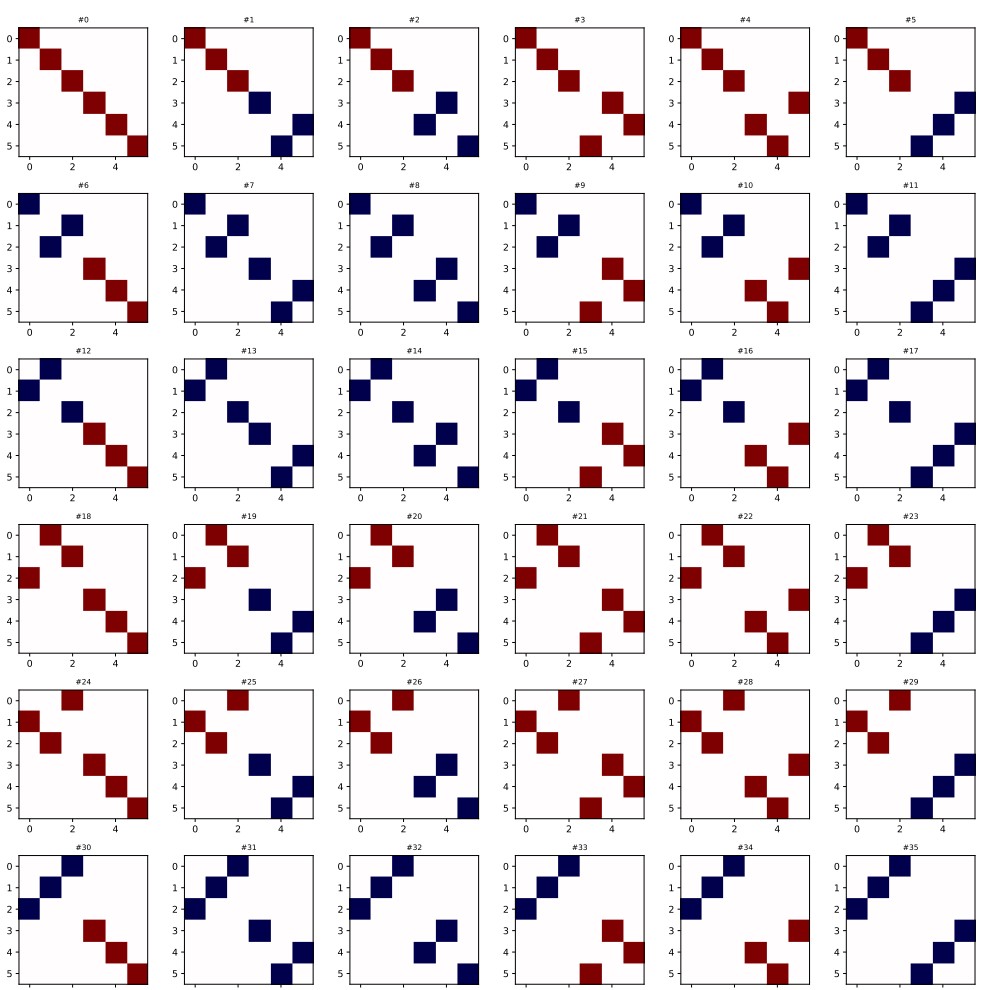

Figure 11: **Permutation - block diagonal with two 3x3 block matrix** Red indicates a cycle sub-block, while blue indicates a non-cycle block.

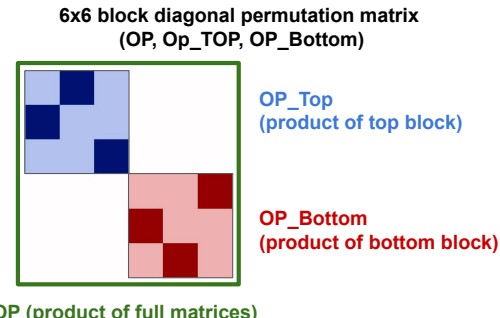

Figure 12: **6x6 block diagonal permutation operations(OP, Op_TOP, OP_Bottom)** OP_Top only acts on the top block matrix and OP_Bottom only acts on the bottom block matrix, while OP is an operation that acts on both the top and bottm blocks.

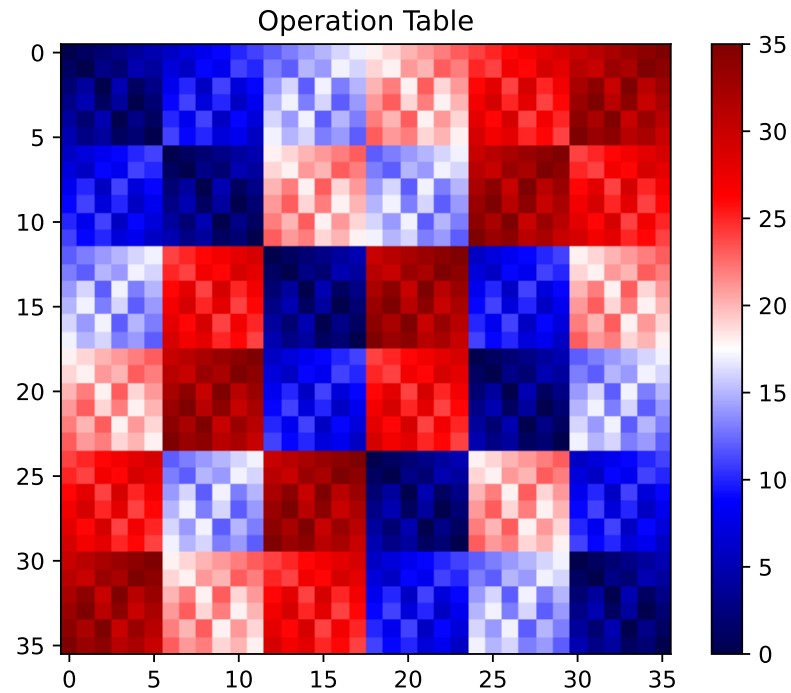

Figure 13: **Operation table (OP) - block diagonal with two 3x3 block matrix**

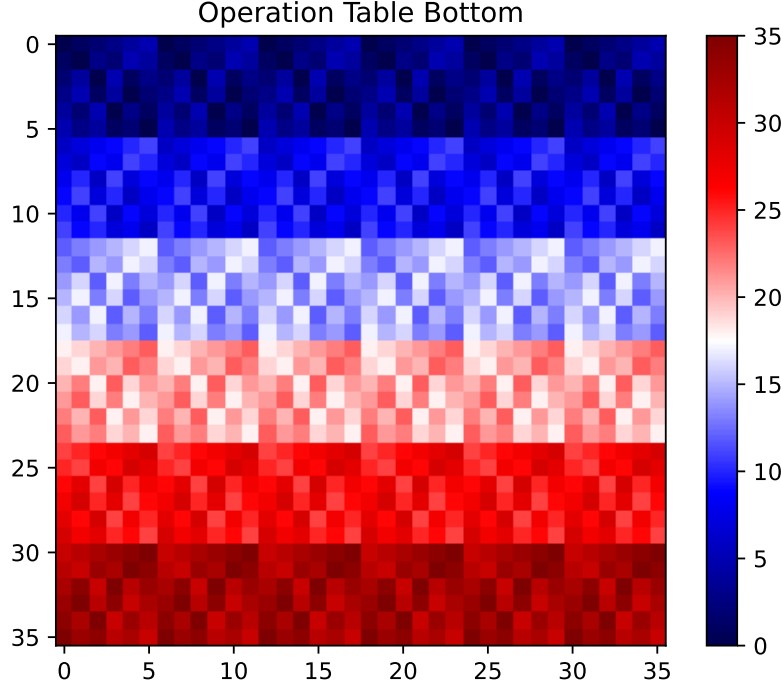

Figure 14: **Operation table (OP_BOTTOM) - block diagonal with two 3x3 block matrix**

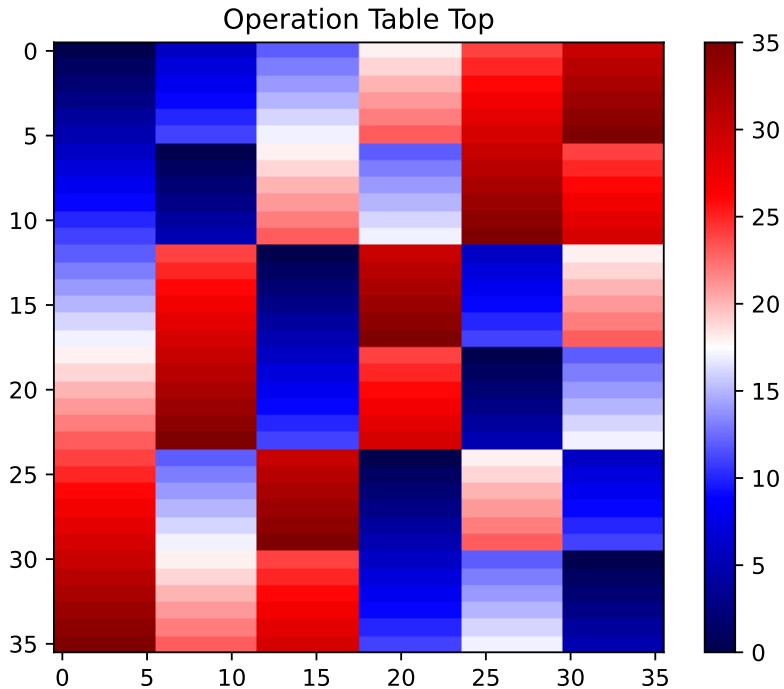

Figure 15: **Operation table (OP_TOP) - block diagonal with two 3x3 block matrix**

## D  SHUFFLED SUM

The Shuffled SUM table was constructed using the following code:

Listing 1: Generate upper triangle and diagonal matrices

```
# Generate upper triangle and diagonal matrices.
# To ensure commutativity i+j=j+i, we will transpose the
# the upper triangle.
upper_triangle_matrix = {(i, j): (i + j) % MOD
    for i in range(MOD)
    for j in range(MOD) if i < j}

# To ensure uniform distribution after shuffling the values,
# we must shuffle the diagonal separately.
# This is because all off-diag r.h.s. are repeated twice
# for commutativity, but not the diagonal entries.
diagonal_matrix = {(i, i): (2 * i) % MOD for i in range(MOD)}

# Function to shuffle values in a dictionary
def shuffle_dict_values(d):
    keys = list(d.keys())
    values = list(d.values())
    random.shuffle(values)
    return dict(zip(keys, values))

# Apply shuffling
shuffled_triangle = shuffle_dict_values(upper_triangle_matrix)
shuffled_diagonal = shuffle_dict_values(diagonal_matrix)
```

As a result, there is no consistent mapping between the original and shuffled values (e.g., the number 1 does not always map to 2), making it difficult for the model to learn a deterministic transformation. While the shuffled sum table remains commutative, we did not enforce associativity. A direct check also confirmed that it does not satisfy the associative property for most triplets.

| A + B = C | Shuffled |
|-----------|----------|
| 0 + 1 = 1 | 0 + 1 = 5 |
| 1 + 0 = 1 | 1 + 0 = 5 |
| 1 + 2 = 3 | 1 + 2 = 1 |
| 5 + 4 = 9 | 5 + 4 = 3 |
| ... | ... |

Figure 16: **Shuffle the summation table.**

## D.1 Shuffled Sum modulo 10

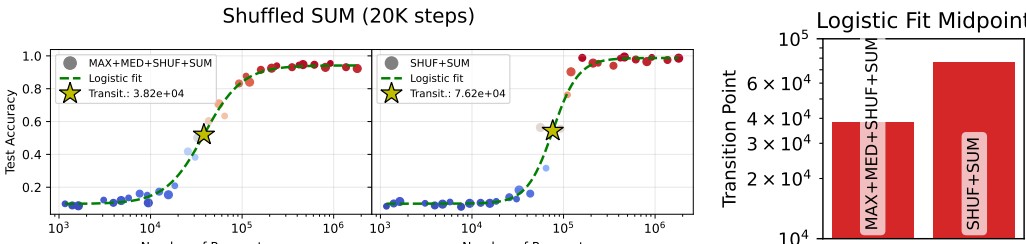

Figure 17: **Shuffled symmetric sum table, Mod 10:** We find that this shuffled version of sum is again more difficult to learn than any of the math operations except for the original Pure SUM, which remains slightly more difficult than even the shuffled version. Additionally, we observe that MAX+MED+Shuffled SUM is again more difficult than all operations except pure sum, suggesting that the number properties played an important role in the other tasks becoming easier in mixed training.

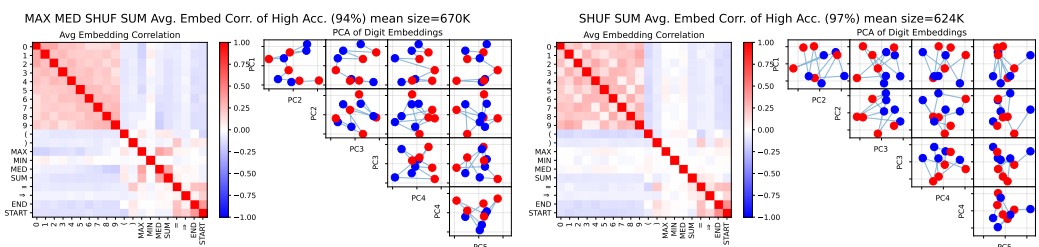

Figure 18: **PCA of embeddings mod 10 for Shuffled symmetric SUM, pure vs mixed with MAX and MED:** The numbers are colored based on parity (odd is red, even is blue). It is not expected that the embeddings show strong signals. There seems to be partial ordering of the numbers, but the clear wave patterns and clear parity separation is not evident.

## D.2 Shuffled Sum modulo 26

We make a symmetric sum table ($A + B = B + A$), with a randomly shuffled right hand side, meaning where if in the table $A + B = C$, $C$ does not the actual arithmetic sum of $A + B$ modulo 26. We do this to test a couple of hypotheses:

1. **Hypothesis 1:** The SUM trained alone is not really learning the logic arithmetic of numbers, but rather memorizing the sum table.

2. **Hypothesis 2:** Joint training with max and med leads to learning number properties.

If the first hypothesis holds, the shuffled sum would also require similarly high number of parameters as the normal sum and show similarly random patterns in the embedding space. If the second hypothesis holds, then joint training of shuffled sum should actually have detrimental effects on learning sum because number properties don't play a role in learning shuffled sum. Thus we expect the jointly trained model to struggle to learn all three operations, or require significantly higher number of parameters to master shuffled sum, compared to joint training on regular sum.

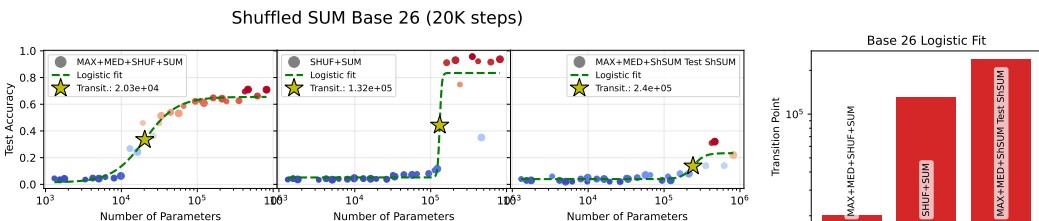

Figure 19: **Shuffled symmetric sum table, Mod 26, 20k steps:** We find that this shuffled version of sum is again more difficult to learn than any of the math operations except for the original Pure SUM, which remains slightly more difficult than even the shuffled version. We also observe that MAX+MED+Shuffled SUM never reaches more than 80% accuracy. The third scatter plot from the let shows the accuracy of the MAX+MED+Shuffled SUM model on the Shuffled SUM test set. We see that the accuracy is very low ($\sim 20\%$ top), showing that the mixed model never learned the shuffled SUM. This may suggest that MAX+MED revealed number properties, but Shuffled SUM was incompatible with those properties, leading to a model that overall cannot solve the two problems (MAX+MED and Shuffled SUM) simultaneously.

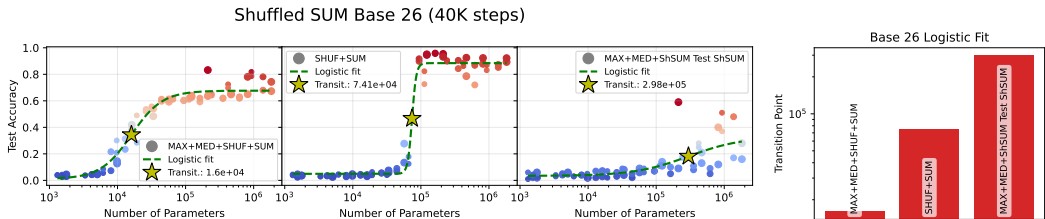

Figure 20: **Shuffled symmetric sum table, Mod 26, 40 steps:** We find that this shuffled version of sum is again more difficult to learn than any of the math operations except for the original Pure SUM, which remains slightly more difficult than even the shuffled version. We also observe that MAX+MED+Shuffled SUM never reaches more than 80% accuracy. The third scatter plot from the let shows the accuracy of the MAX+MED+Shuffled SUM model on the Shuffled SUM test set. We see that the accuracy is very low ($\sim 20\%$ top), showing that the mixed model never learned the shuffled SUM. This may suggest that MAX+MED revealed number properties, but Shuffled SUM was incompatible with those properties, leading to a model that overall cannot solve the two problems (MAX+MED and Shuffled SUM) simultaneously.

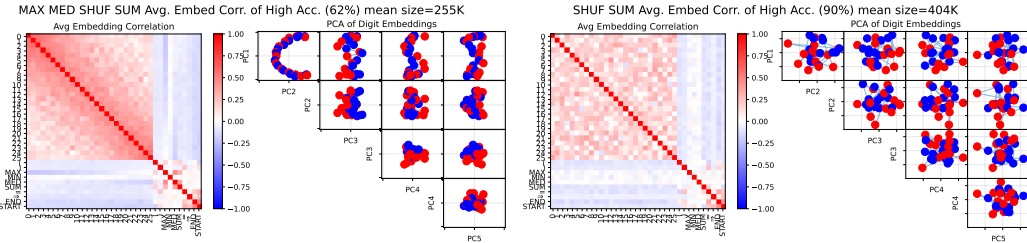

Figure 21: **PCA of embeddings for Shuffled symmetric SUM, pure vs mixed with MAX and MED, Mod 26:** The numbers are colored based on parity (odd is red, even is blue). There seems to be partial ordering of the numbers, but the clear wave patterns are not visible. We do almost observe parity separation in PC3 and PC4, albeit with some noise. It is curious that the system still learns partial parity, but evidently this feature did not allow the system to learn the shuffled SUM perfectly.

## E  DEEP TRANSFORMER VS. RECURRENT TRANSFORMER

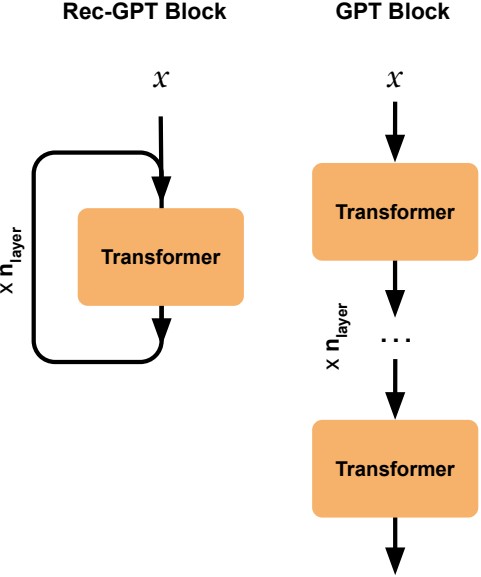

Figure 22: **Deep transformer vs. Recurrent Transformer** In the Rec-GPT block, there is only one transformer block that is applied recursively *n* times, so each output becomes the input for the next step. In contrast, in a traditional GPT block, there are *n* transformer blocks stacked sequentially on top of each other.

## F  RESULTS USING DEEP TRANSFORMER BASED MODELS

### F.1  EXPERIMENT SETTINGS

1. **Model**: GPT model (Karpathy (2022))
2. **Number of layers**: 1, 2, 3, 4, 5, 6
3. **Number of head**: 1
4. **Embedding dimension**: 4, 8, 12, 16, 24, 32, 48, 64, 96, 128, 192, 256
5. **Number of random seed**: 5
6. **Context window size**: 128
7. **Batch size**: 64
8. **Optimizer**: Adam
9. **Minimum learning rate**: 1e-4
10. **Maximum interation steps**: 20000/50000
11. **Early Stopping criteria**: Early stopping was applied after 2000 iterations when the change in training loss was below $\Delta_{min}$ = 2.5e-4 for 10 consecutive evaluation steps.
12. **Data** We generate 50,000 initial equations and split them into training and test sets by excluding 100 randomly selected triplets. The training set consists of approximately 45,000 examples that do not contain any of the excluded triplets. The test set comprises around 2,000 examples, carefully curated to ensure that the excluded triplets do not appear, even in the final step of each equation. We evaluate model performance on a subset of 1,000 final test examples.
13. **Vocab - Base 10 ListOps**: `%()+-/0123456789=>es`
14. **Vocab - Base 26 ListOps**: `ABCDEFGHIJKLMNOPQRSTUVWXYZse()+-/%=>`

15. **Hardware**: All simulations were run on a mix of GPUs, including NVIDIA A100, H200, RTX 4090, and A30, as well as on a standard modern laptop. The experiments are lightweight, requiring approximately 1.5GB of RAM, and can be executed efficiently on any recent laptop-class device.

## F.2    EXPERIMENTS ON LISTOPS.

1. **Base 10**: - no CoT/CoT: MAX, MED, SUM, MAX/MED/SUM

2. **Base 10**: - all combination: MAX, MIN, MED, SUM + MAX/MED/SUM-RANDOM SHUFFLED

3. **Base 26**: - all combination: MAX, MIN, MED, SUM + MAX/MED/SUM-RANDOM SHUFFLED

4. **Triplet**: MAX+MED+SUM 2700 (2700 training data samples = 900 MAX + 900 MED + 900 SUM), MAX+MED+SUM 900 (900 training data samples = 300 MAX + 300 MED + 300 SUM), SUM (900 training data samples)

## F.3    DATA AND PROCESSING

### F.3.1    DATASET NOTATION

ListOps consists of nested mathematical equations involving operations such as min, max, median, and sum modulo 10 applied to single-digit numbers (0-9). It uses the Polish notation: (operation, [inputs]) For example: `max(3,min(7,4,9))=4`.

$$\texttt{max(3,min(7,4,9))=4} \quad \Rightarrow \quad \textbf{Polish:} \quad \texttt{(max,3,(min,7,4,9))=4}$$

To disentangle any complexity arising from tokenization we further simplify these expression by representing the by symbols: '+' for max, '−' for min, '/' for median, and '%' for sum modulo 10. For example:

$$\texttt{(max,3,(min,7,4,9))=4} \quad \Rightarrow \quad \textbf{Our notation:} \quad \texttt{s(+3(-749))=4e}$$

In this notation, 's' denotes the start of the expression, 'e' marks the end, and parentheses indicate nesting levels.

### F.3.2    TOKENIZATION

We employ a character-based tokenization strategy for processing ListOps expressions. This approach offers several advantages:

1. **Simplicity:** Character-level tokenization eliminates the need for complex tokenization rules or a large vocabulary.

2. **Generalizability:** It allows the model to potentially generalize to unseen number combinations or deeper nesting levels.

Each character in the ListOps expression, including digits, operation symbols, and structural elements (parentheses, 's', 'e'), is treated as a separate token. This granular representation enables the model to learn the syntactic structure of the expressions alongside their semantic content.

### F.3.3    CHAIN OF THOUGHT IMPLEMENTATION

We find that directly solving nested ListOps in one step can be quite challenging for transformer model (Fig. 23) Even with a **maximum of three nesting levels with three operands (inputs)** we find that GPT models with over 10 million parameters still fail to learn the task. To enhance model performance, particularly on more complex operations like sum modulo 10, we introduced a chain of thought (CoT) approach in our training data. This method involves providing step-by-step solutions that resolve the deepest nesting level at each step. For example:

$$\texttt{s(\%12(\%34))>(\%127)>0=0e}$$

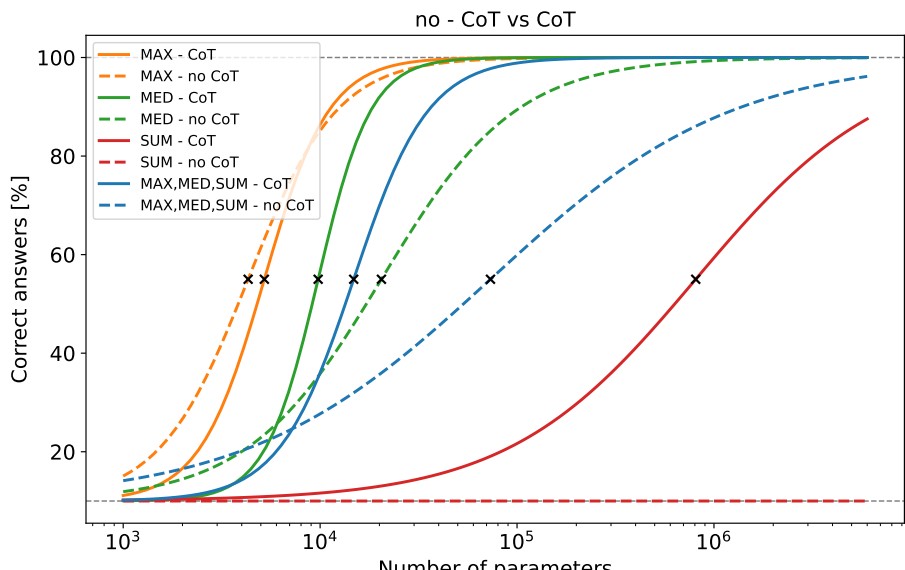

Figure 23: **no - CoT vs. CoT.** Providing solutions as chain-of-thought (CoT) helps models learn the tasks. In almost all cases, CoT accelerates learning, enabling smaller models to succeed. This effect is especially strong for the sum operation, which cannot be learned without CoT.

In this CoT representation:

- The initial expression is `s(%12(%34))`
- The first step resolves the innermost operation: `(%34)` becomes '7'
- The intermediate result is shown: `s(%12(7))>(%127)`
- The process continues until the final result is reached: `s(%12(7))>(%127)>0=0e`

This CoT approach serves multiple purposes: 1. It guides the model through the problem-solving process, mimicking human-like reasoning. 2. It provides more granular supervision, potentially aiding in learning complex operations. 3. It allows us to study how models learn to break down and solve nested problems. Our experiments show that this CoT method significantly improves model performance, particularly for the challenging sum modulo 10 operation (Fig. 23).

### F.4 MODULO 26

We define the following token vocabulary `ABCDEFGHIJKLMNOPQRSTUVWXYZse()+-/%=>`, where letters are mapped to integers such that $A \to 0$, $B \to 1$, ..., $Z \to 25$.

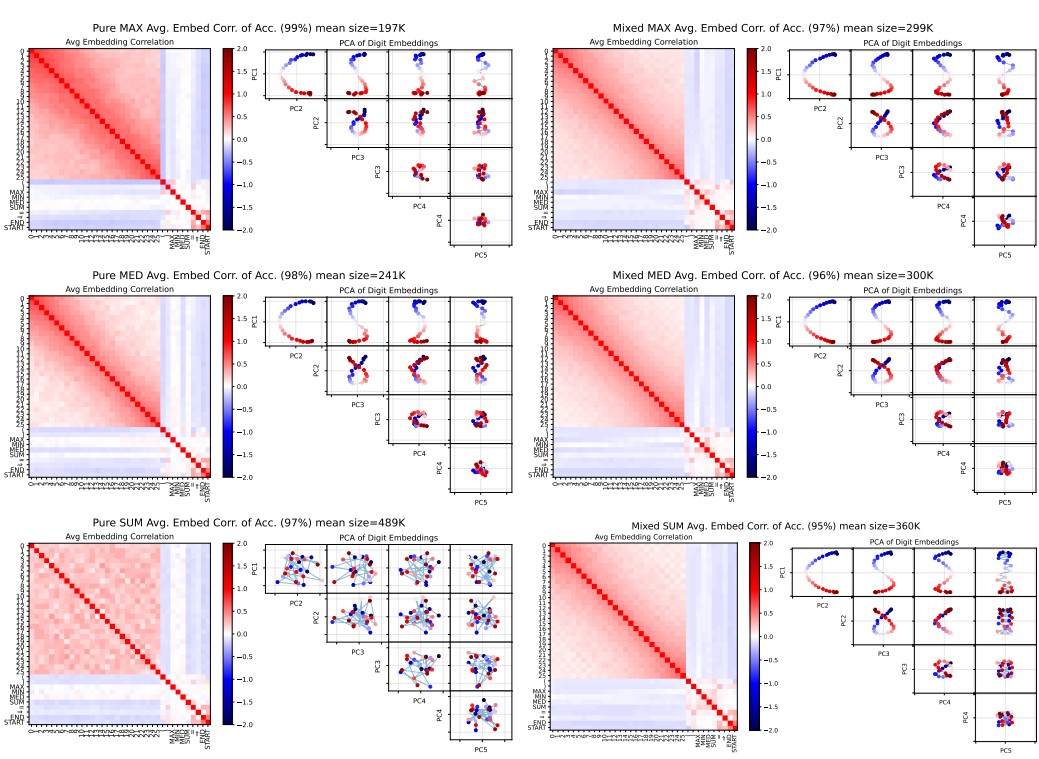

Figure 24: **PCA of embeddings:** We choose all models which reached over 95% test accuracy. Each row shows the average correlation matrix and top PCs for models trained on either a single operation, e.g. Pure+MAX, or all mixtures involving a given operation, e.g. Mixed+MAX. Again, pure SUM does not show a discernible structure in the embeddings, whereas all cases do.

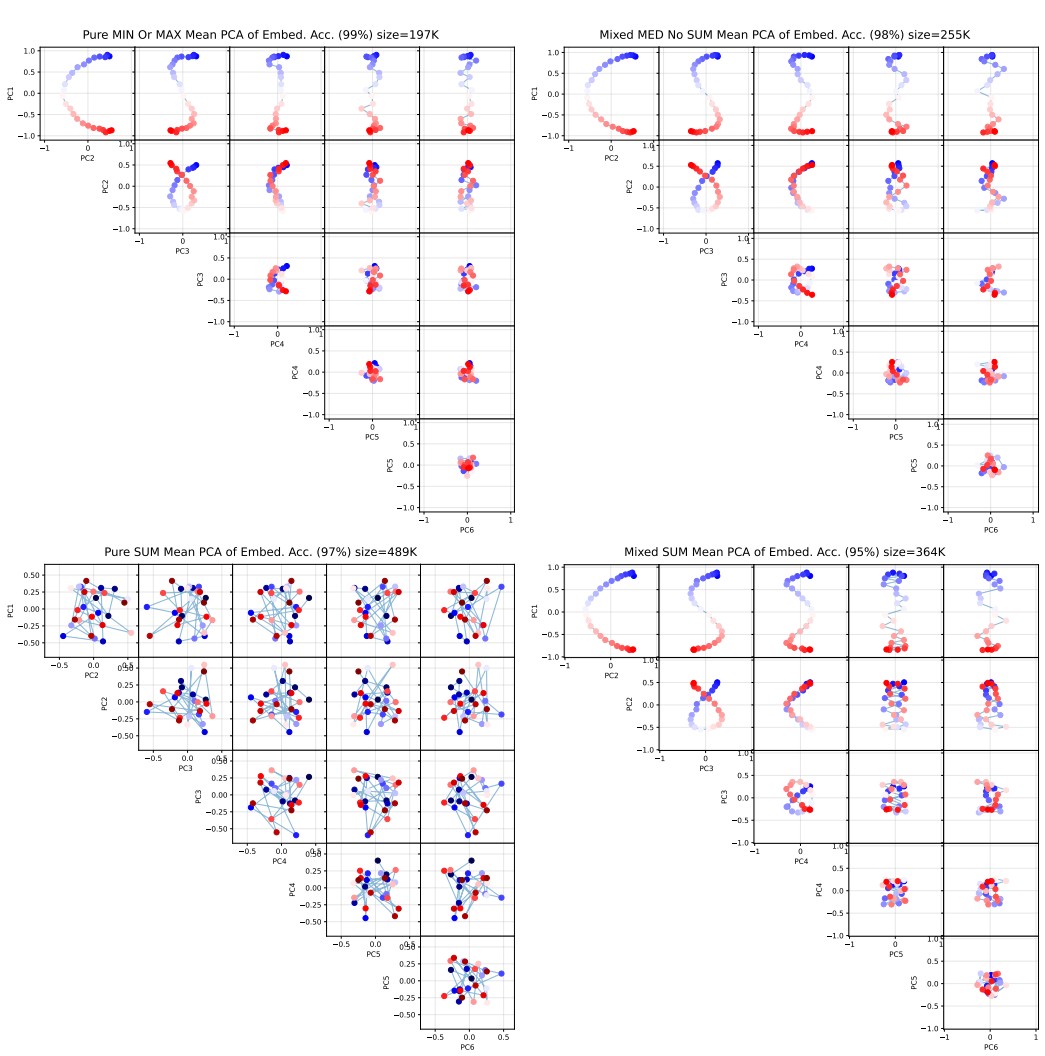

Figure 25: **PCA of embeddings:** We choose all models which reached over 95% test accuracy. Each row shows the average correlation matrix and top PCs for models trained on either a single operation, e.g. Pure+MAX, or all mixtures involving a given operation, e.g. Mixed+MAX. Again, pure SUM does not show a discernible structure in the embeddings, whereas all cases do.

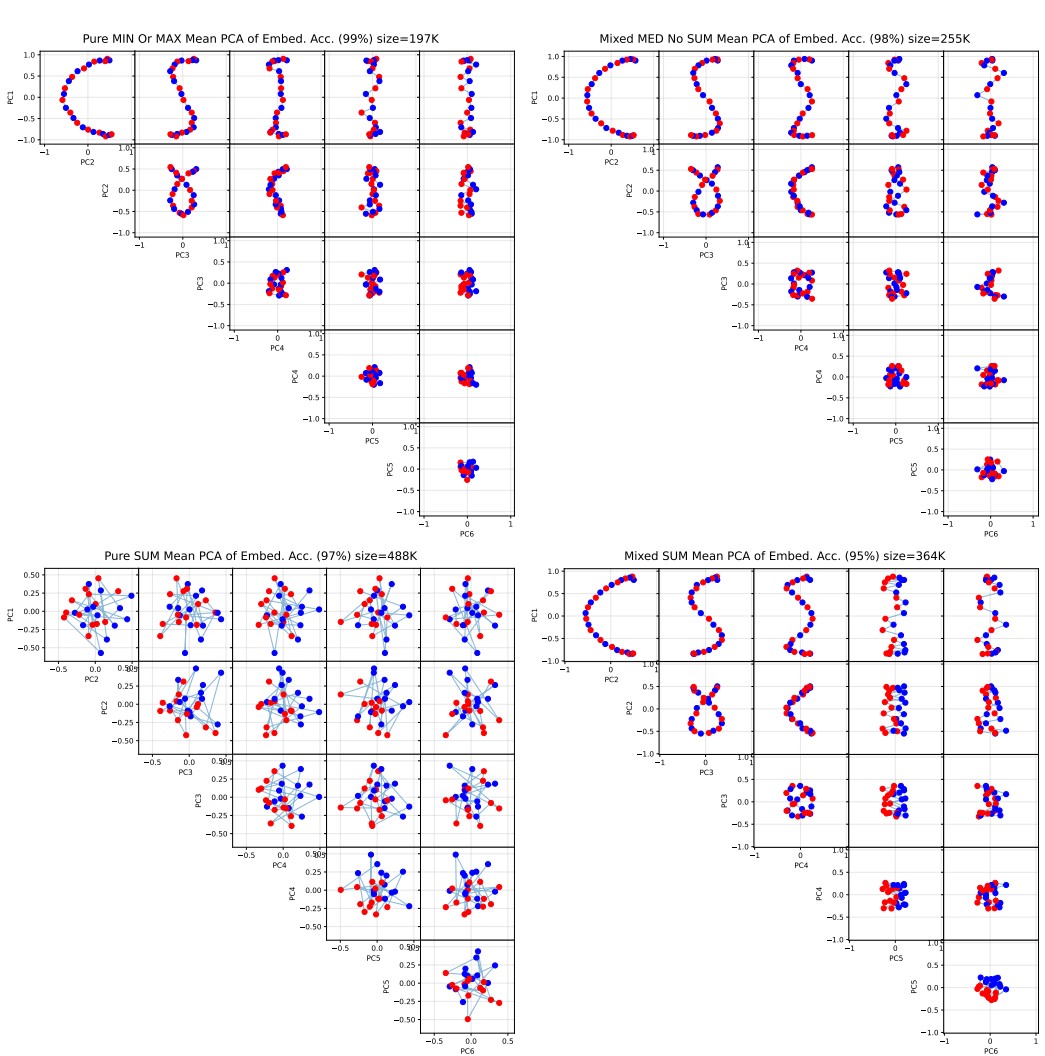

Figure 26: **PCA of embeddings Odd vs Even:** Same plot as above, only odd numbers colored red and even colored blue. Mixed SUM shows a clear odd-even separation in a few of the top PCs. Such a separation is not clearly observed in other cases. Interestingly, Pure SUM approximately separates odd-even, suggesting such separation may play a role in its algorithm.

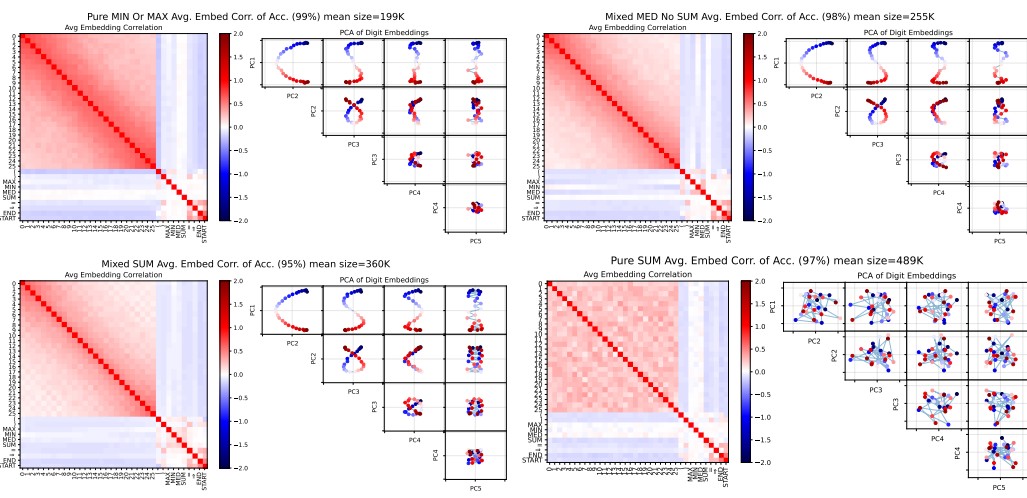

Figure 27: **PCA of embeddings:** We choose all models which reached over 90% test accuracy. Each row shows the average correlation matrix and top PCs for models trained on either a single operation, e.g. Pure MAX, or all mixtures involving a given operation, e.g. Mixed SUM. Interestingly, pure SUM does not show a discernible structure in the embeddings, whereas all other cases do. Notably, Mixed SUM models exhibit a prominent odd-even separation in PC5.

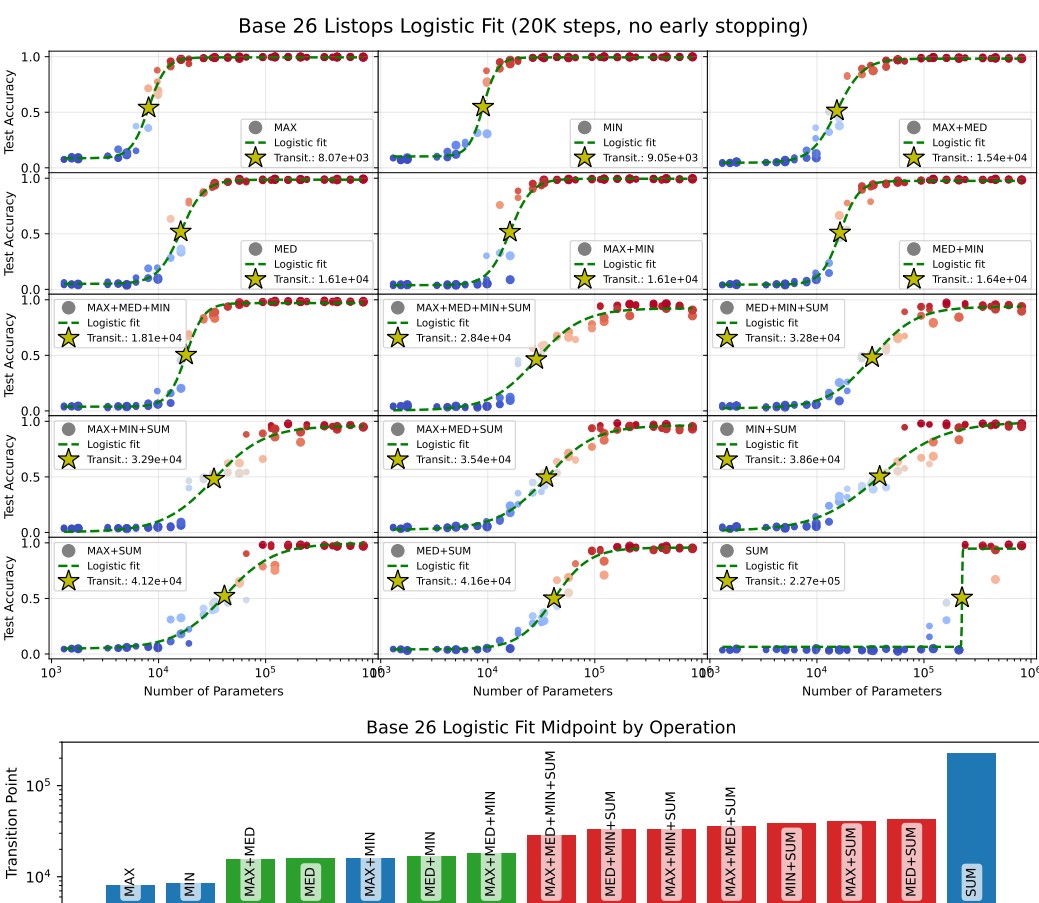

Figure 28: **Emergence of abilities in ListOps:** Each plot shows the same group of small transformer models trained on a different mix of the four operations MAX, MIN, MED, and SUM. Red dots are models reaching more than 50% accuracy, and blue dots are less than 50%. The dashed green line is a logistic fit, and the yellow star indicates the transition point at 50%. The x-axis is the model size (number of parameters), and the plots are sorted in ascending order of transition points. The bottom panel shows a bar plot of the model sizes at the transition points, with each group distinguished by a different color.

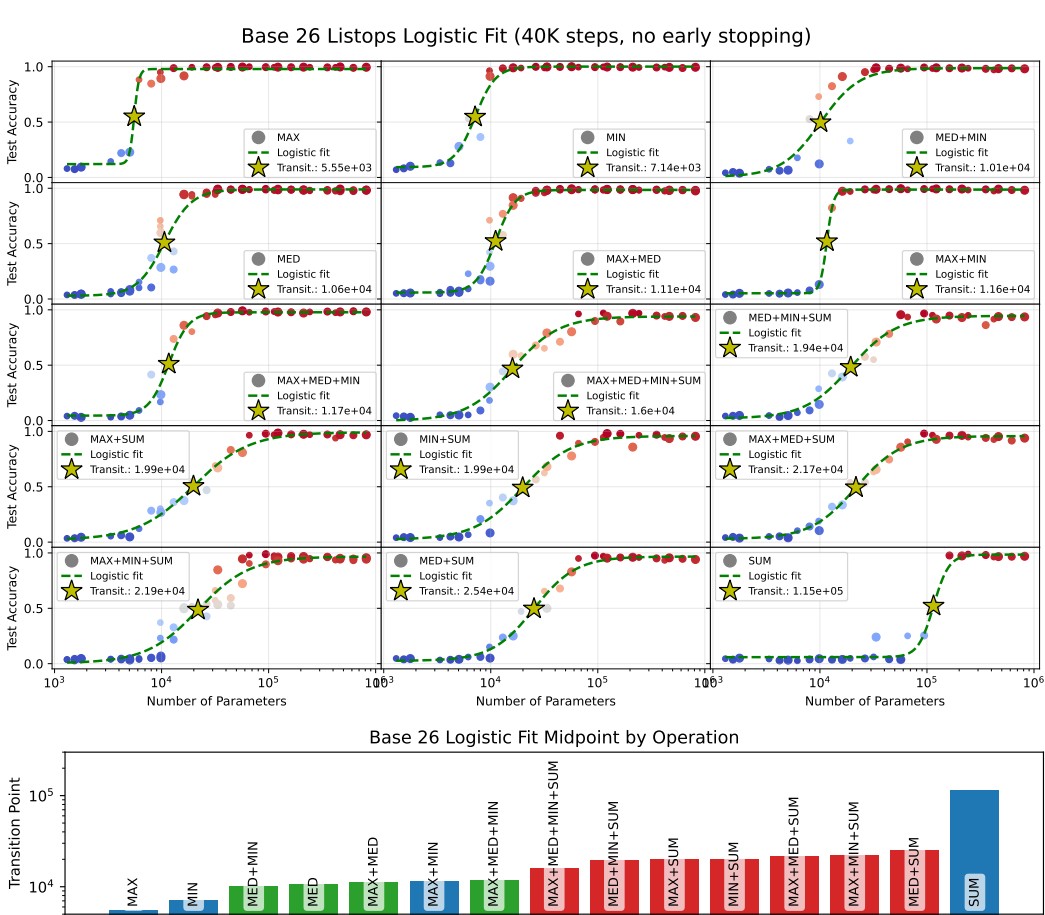

Figure 29: **Emergence of abilities in ListOps, 40k steps:** Long training. Each plot shows the same group of small transformer models trained on a different mix of the four operations MAX, MIN, MED, and SUM. Red dots are models reaching more than 50% accuracy, and blue dots are less than 50%. The dashed green line is a logistic fit, and the yellow star indicates the transition point at 50%. The x-axis is the model size (number of parameters), and the plots are sorted in ascending order of transition points. The bottom panel shows a bar plot of the model sizes at the transition points, with each group distinguished by a different color.

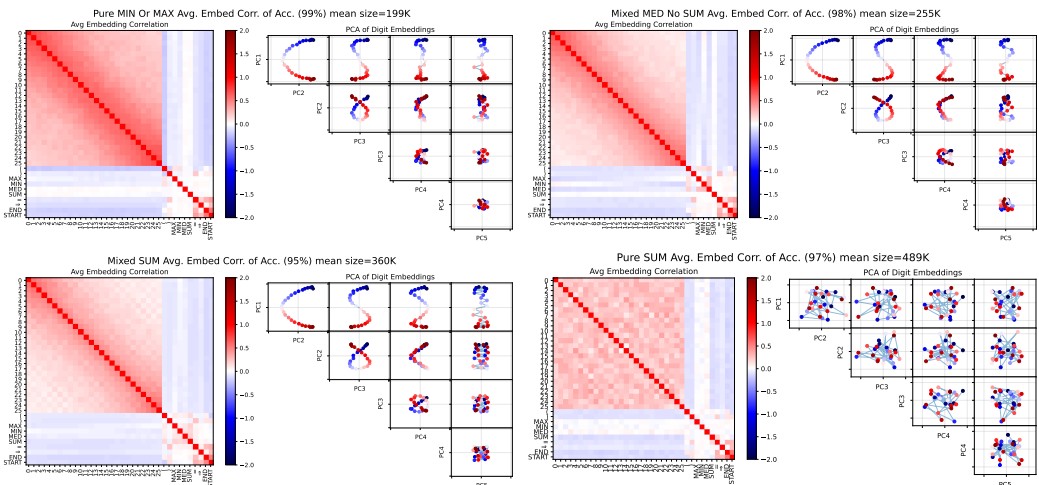

Figure 30: **PCA of embeddings:** We choose all models which reached over 90% test accuracy. Each row shows the average correlation matrix and top PCs for models trained on either a single operation, e.g. Pure MAX, or all mixtures involving a given operation, e.g. Mixed SUM. Interestingly, pure SUM does not show a discernible structure in the embeddings, whereas all other cases do. Notably, Mixed SUM models exhibit a prominent odd-even separation in PC5.

## F.5 MODULO 10

We conducted the same experiments also on mod 10. The smaller number of numbers makes definitive statements about some of the patterns more challenging. But all the patterns we observed in mod 26 also have parallels in mod 10, including the prominent odd-even split for operations involving SUM. Token vocabulary: `%()+-/0123456789=>es`

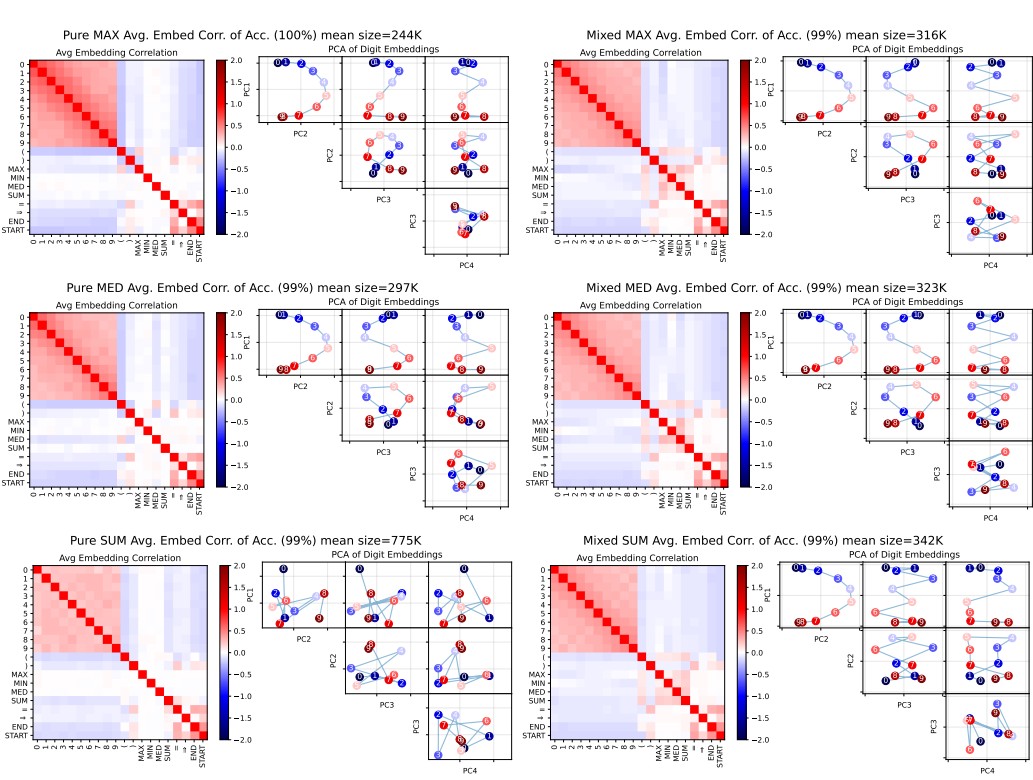

Figure 31: **PCA of embeddings:** We choose all models which reached over 95% test accuracy. Each row shows the average correlation matrix and top PCs for models trained on either a single operation, e.g. Pure+MAX, or all mixtures involving a given operation, e.g. Mixed+MAX. Again, pure SUM does not show a discernible structure in the embeddings, whereas all cases do.

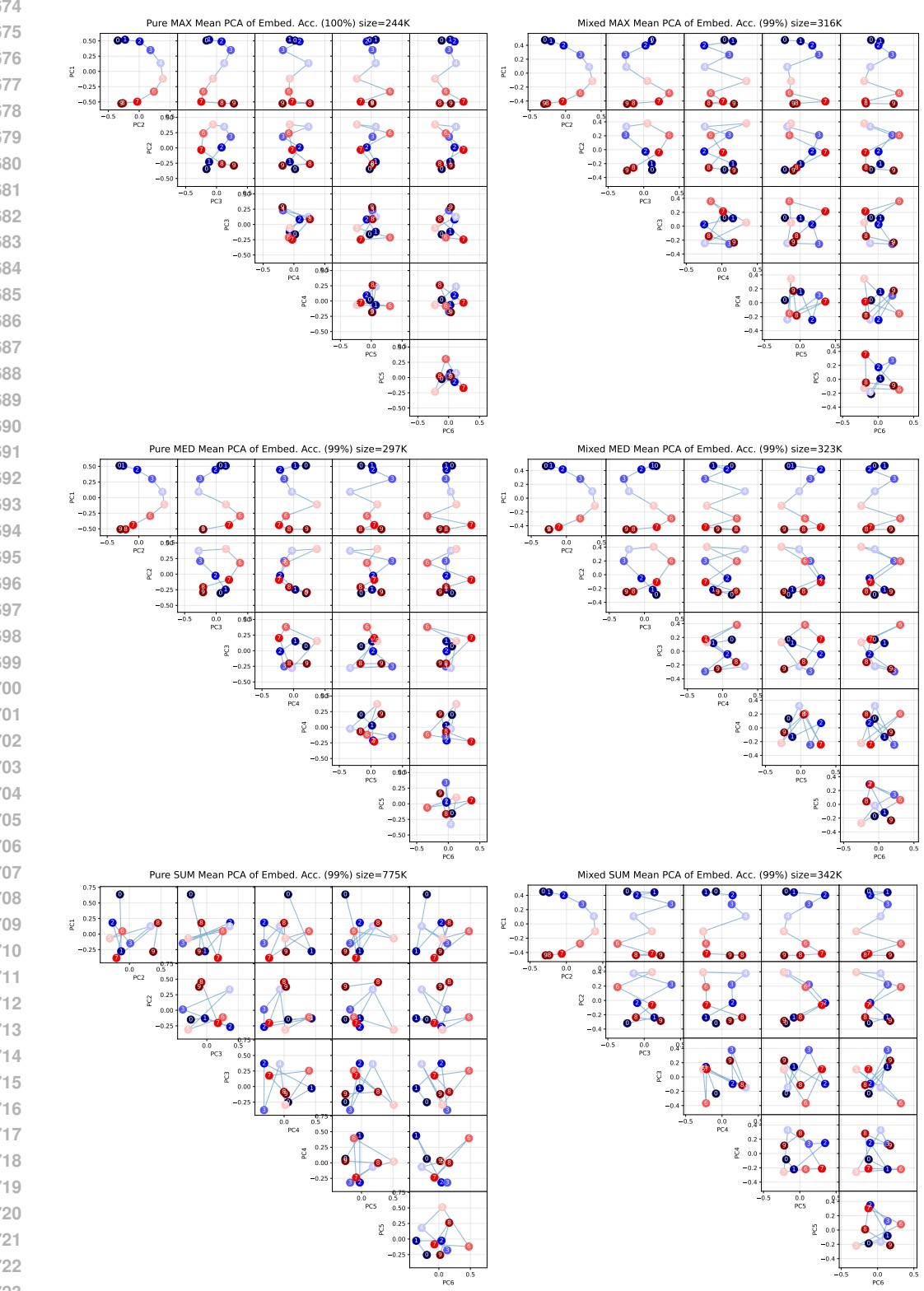

Figure 32: **PCA of embeddings:** We choose all models which reached over 95% test accuracy. Each row shows the average correlation matrix and top PCs for models trained on either a single operation, e.g. Pure+MAX, or all mixtures involving a given operation, e.g. Mixed+MAX. Again, pure SUM does not show a discernible structure in the embeddings, whereas all cases do.

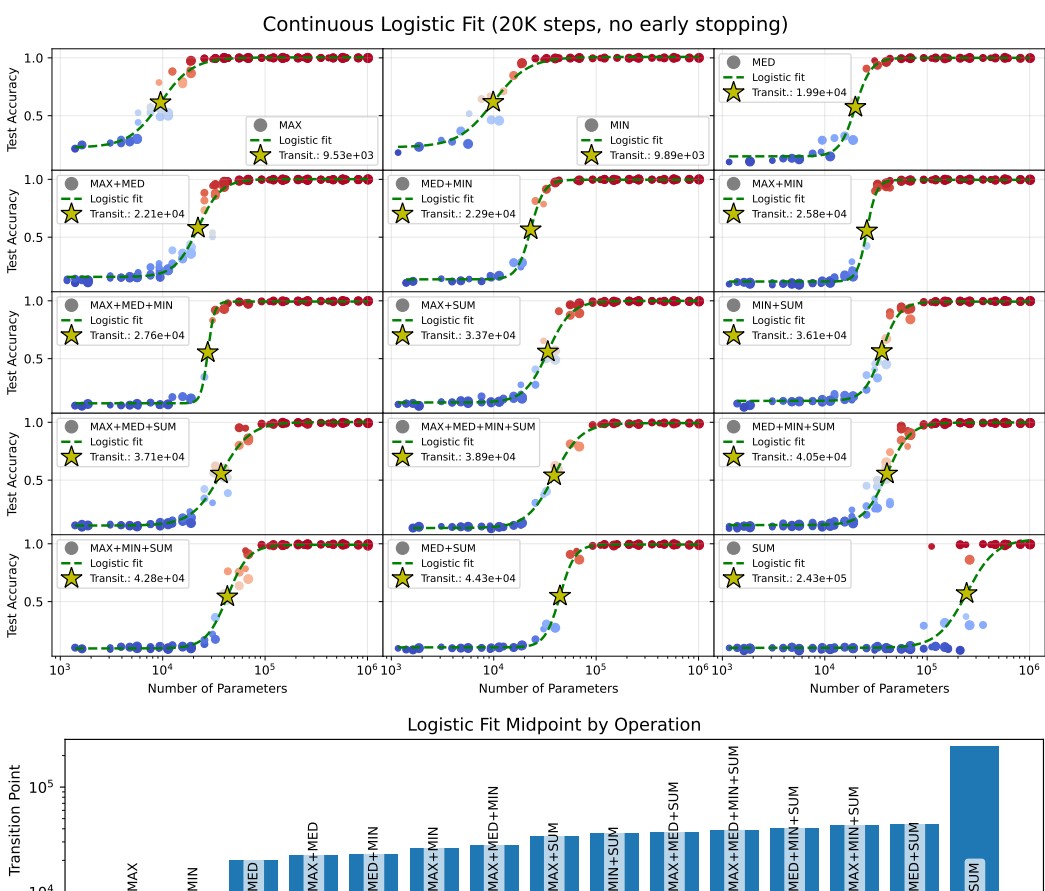

Figure 33: **Emergence of of abilities in ListOps, Mod 10:** Each plot shows the same group of small transformer models trained on different variants of ListOps. Each variant uses a different mix of the four operations MAX, MIN, MED, SUM. Red dots are model reaching more than 50% accuracy, and blue are less than 50%. The dashed green line is a logistic fit and the yellow star indicates the transition point at 50%. The x axis is the model size and the plots are sorted in ascending order of transition points. The bottom is a bar plot showing the model size at the transition point. We observe that SUM is a clear outlier, with models requiring significantly more parameters to learn SUM. Surprisingly, combining SUM with other operations dramatically reduces the transition point, with model less than half the size easily reaching 100% accuracy.

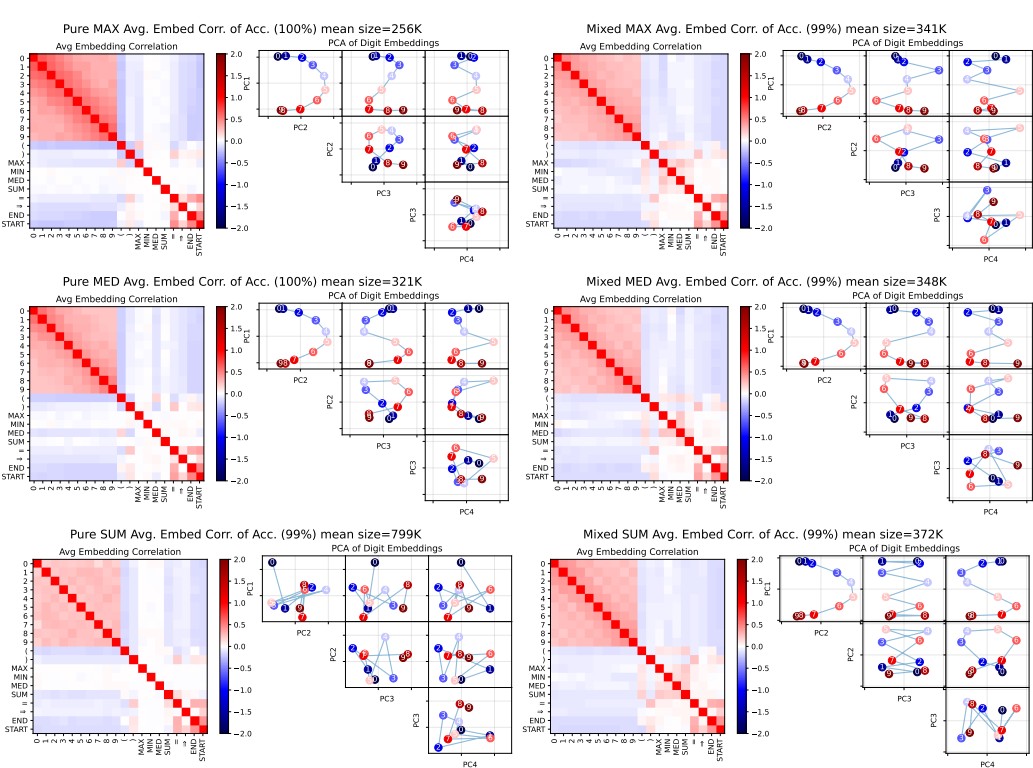

Figure 34: **PCA of embeddings with the same sample size (67):** We choose the smallest 67 models which reached over 95% test accuracy. Each row shows the average correlation matrix and top PCs for models trained on either a single operation, e.g. Pure+MAX, or all mixtures involving a given operation, e.g. Mixed+MAX. We observe that the PCs get slightly distorted with the smaller sample size compared to fig. 32, but the overall structure stays the same. Again, pure SUM does not show a discernible structure in the embeddings, whereas all cases do.

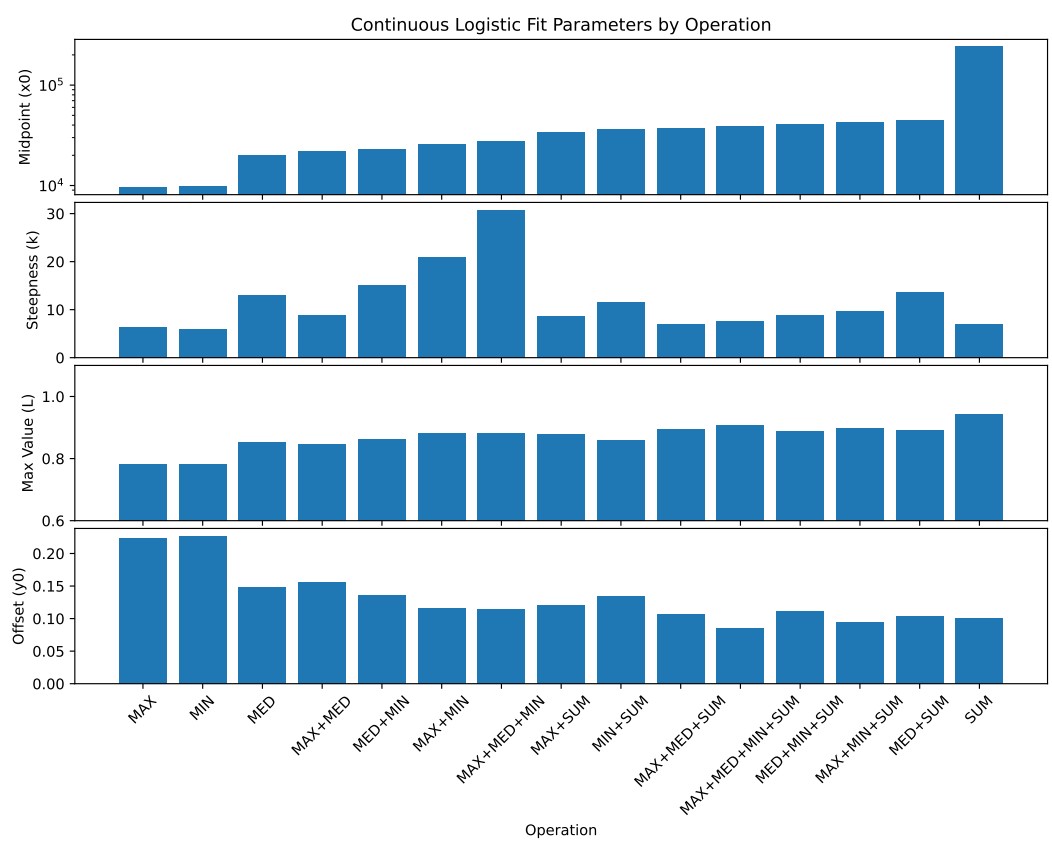

Figure 35: **Parameters of continuous logistic regression fitting accuracy to number of parameters (No Early stopping)**.

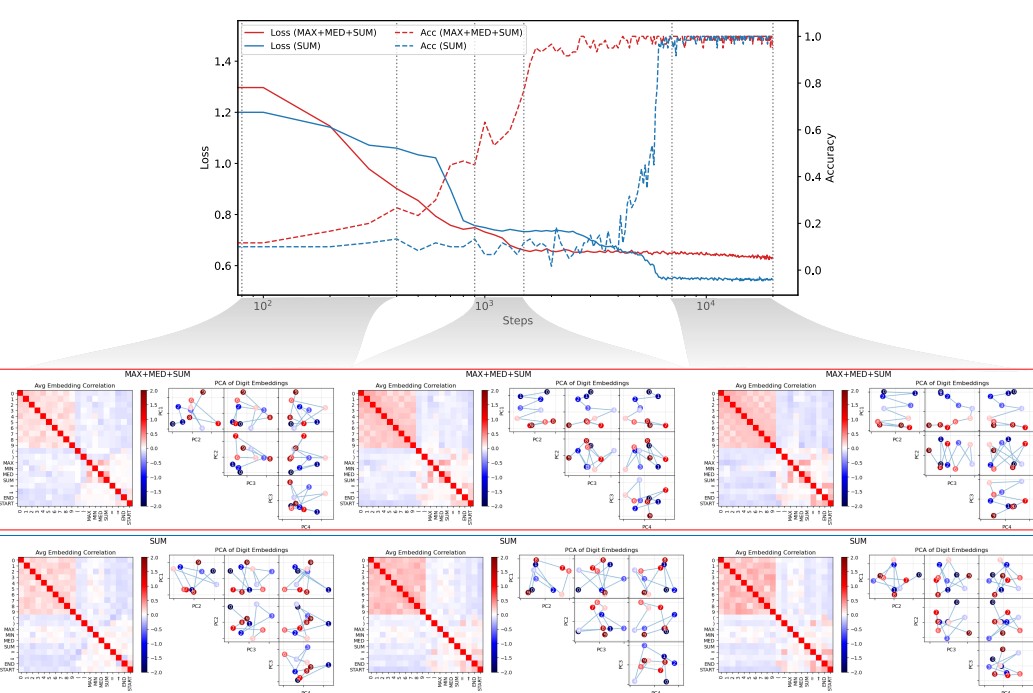

Figure 36: **Evolution of training loss, accuracy, and principal components of cosine similarities in the embedding layer. Modulo 10 ListOps.** The top main figure shows the evolution of training loss (solid lines) and test accuracy (dashed lines) for models with an embedding dimension of 128 and 3 layers, trained either on SUM-only data (blue) or on mixed MAX+MED+SUM data (red). Curves represent the mean across three independent runs, with shaded regions indicating one standard deviation. All models were trained for 20000 iterations. The red and blue boxes beneath the main plot display the average embedding representations at different training stages (indicated by vertical gray dashed lines). PCA reveals that models trained on MAX+MED+SUM data progressively develop a structured representation of numerical concepts, accompanied by a steady decrease in loss. In contrast, models trained solely on SUM data exhibit no clear structure in the embedding space and show long plateaus in the loss curve. This suggests that discovering an effective algorithm for the SUM operation in isolation requires significantly more exploration during training, in contrast to the more efficient joint training setting.

### F.6 TRIPLET EXPERIMENTS

To increase control in our experimental setup, we train models only on triplet inputs 37. We construct 1000 unique triplet samples, splitting them into 900 for training and 100 for testing.

For joint training on the MAX, MED, and SUM operations, we use a combined training set of 2700 triplets (900 per operation) and evaluate performance on the same 100 excluded SUM samples. For a fair comparison, we also create a balanced training set of 900 samples by selecting 300 examples from each of MAX, MED, and SUM.

With early stopping, models trained on the full 2700-sample mixed dataset achieve 100% accuracy on the SUM test set, while those trained on only 900 SUM samples reach approximately 50% accuracy. When trained for 50k steps without early stopping, models eventually learn the task, but SUM requires significantly more parameters—consistent with earlier observations. These results highlight that learning the SUM operation in isolation is more challenging and slower (Fig. 37).

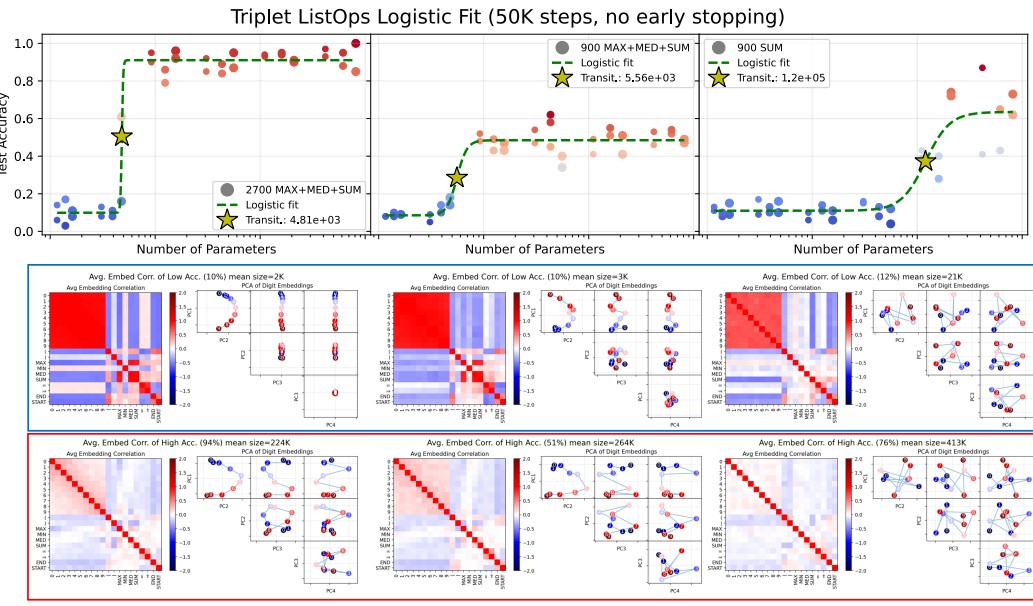

Figure 37: **Training only on triplet ListOps data.** The first row shows the same group of small transformer models trained on different variants of the triplet dataset. Each training set is constructed from 900 unique triplets. The 2700 MAX+MED+SUM dataset includes all 900 triplets, each labeled with three different operations. The 900 MAX+MED+SUM dataset contains 300 examples per operation, randomly sampled from the 2700 set. The 900 SUM dataset contains only the 900 unique SUM triplets. In all cases, the test set comprises the same held-out 100 SUM triplets. Red dots indicate models that exceed 50% test accuracy, while blue dots denote models that fall below this threshold. The dashed green line represents a logistic fit, and the yellow star marks the transition point at 50% accuracy. The x-axis shows the model size (in number of parameters), and subplots are ordered by increasing transition threshold. The red and blue boxes beneath the main plot display the average PCA embedding representations at selected regions; red corresponds to high-accuracy models, while blue indicates low-accuracy ones. Models trained on the combined MAX+MED+SUM task develop a discernible structure during training, unlike models trained solely on SUM. Notably, augmenting MAX and MED data with only 300 SUM samples allows the model to achieve 50% accuracy on a held-out 100 SUM sample test set, using a model 10× smaller than those trained on 900 SUM samples alone, which only reach $\approx 80\%$ accuracy.

## G OBSERVATIONS FROM THE NORM OF ATTENTION AND FEEDFORWARD OUTPUTS

To investigate the internal dynamics of our models, we focused on the final layer of a 3-layer transformer network, featuring a single attention head and an embedding dimension of 128. Our analysis centered on comparing the behavior of models trained on "ALL3" operations (MAX+MED+SUM) versus those trained solely on the "SUM" operation.

We introduced a novel metric to quantify the impact of different components within the network: the ratio of output to input norms for both the self-attention (SA) and feedforward (FFN) sublayers. Specifically, we computed:

1. Attention ratio: $r_{attn} = \frac{\|SA(LN_1(x))\|}{\|x\|}$

2. Feedforward ratio: $r_{ffwd} = \frac{\|FFN(LN_2(x_1))\|}{\|x_1\|}$

where $LN_1$ and $LN_2$ are layer normalization operations, and $x_1$ is the output of the self-attention sublayer. These ratios provide insight into how much each component modifies its input, serving as a proxy for the component's impact on the overall computation.

We analyzed the distribution of these ratios across a test set consisting of sum operations for both the 'ALL3' and 'SUM' models. Kernel Density Estimation (KDE) plots were used to visualize the distributions, and we employed several statistical measures to quantify the differences.

**Attention Sublayer** The attention sublayer showed moderate but statistically significant differences between the 'ALL3' and 'SUM' models:

- Kolmogorov-Smirnov test: statistic = 0.1592, p-value < 0.0001
- Jensen-Shannon divergence: 0.1591
- Wasserstein distance: 0.0696
- Effect size (Cohen's d): 0.1761
- 95% CI for mean difference: (0.0466, 0.0860)

The KDE plot revealed that the 'ALL3' model's attention ratio distribution was more concentrated and peaked higher than the 'SUM' model's distribution. The positive effect size and confidence interval indicate that the 'ALL3' model generally had higher attention ratios.

**Feedforward Sublayer** The feedforward sublayer exhibited more pronounced differences:

- Kolmogorov-Smirnov test: statistic = 0.2461, p-value < 0.0001
- Jensen-Shannon divergence: 0.1617
- Wasserstein distance: 0.2830
- Effect size (Cohen's d): -0.3379
- 95% CI for mean difference: (-0.3042, -0.2226)

The KDE plot for the feedforward ratios showed a clear shift between the two distributions. The 'SUM' model's distribution was shifted towards higher values, as confirmed by the negative effect size and confidence interval.

**Interpretation** These results reveal distinct operational patterns between models trained on 'ALL3' operations versus those trained solely on 'SUM':

1. In the attention sublayer, the 'ALL3' model shows slightly higher ratio values, **suggesting that attention mechanisms play a more pronounced role when the model is trained on diverse operations.** This could indicate that the attention sublayer is capturing more complex patterns or relationships necessary for handling multiple operations.

2. Conversely, in the feedforward sublayer, the 'SUM' model demonstrates significantly higher ratio values. This **suggests that when trained on 'Sum' alone, the model relies more heavily on the feedforward network for computation.** This could imply that the 'SUM' operation is being implemented more directly through feedforward transformations.

3. The larger effect size in the feedforward layer (-0.3379) compared to the attention layer (0.1761) indicates that the difference in behavior is more pronounced in the feedforward component.

These observations suggest a trade-off in how the network allocates its computational resources. The 'ALL3' model appears to leverage its attention mechanism more, potentially to handle the diversity of operations it was trained on. In contrast, the 'SUM' model seems to channel more of its computation through the feedforward network, possibly developing a more specialized but less flexible approach to solving the sum operation.

This analysis provides evidence that the internal dynamics of transformer models adapt significantly based on the diversity of tasks they are trained on, even when evaluated on the same type of operation ('SUM'). It highlights the importance of considering task diversity in understanding and optimizing neural network architectures.

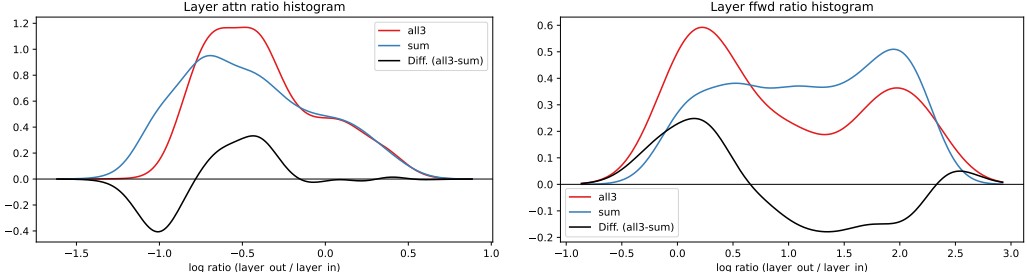

Figure 38: **Comparison of layer output/input ratio distributions** for models trained on all three operations (all3 - MAX+MED+SUM) versus sum operation alone (SUM). **Left:** Attention layer ratio histogram. **Right:** Feedforward layer ratio histogram. The x-axis represents the log ratio of layer output norm to input norm, while the y-axis shows the density. The black line represents the difference between the all3 and sum distributions (all3 - sum). These plots illustrate distinct operational patterns between the two models, with the attention layer showing increased activity in the all3 model and the feedforward layer demonstrating higher ratios in the sum model.

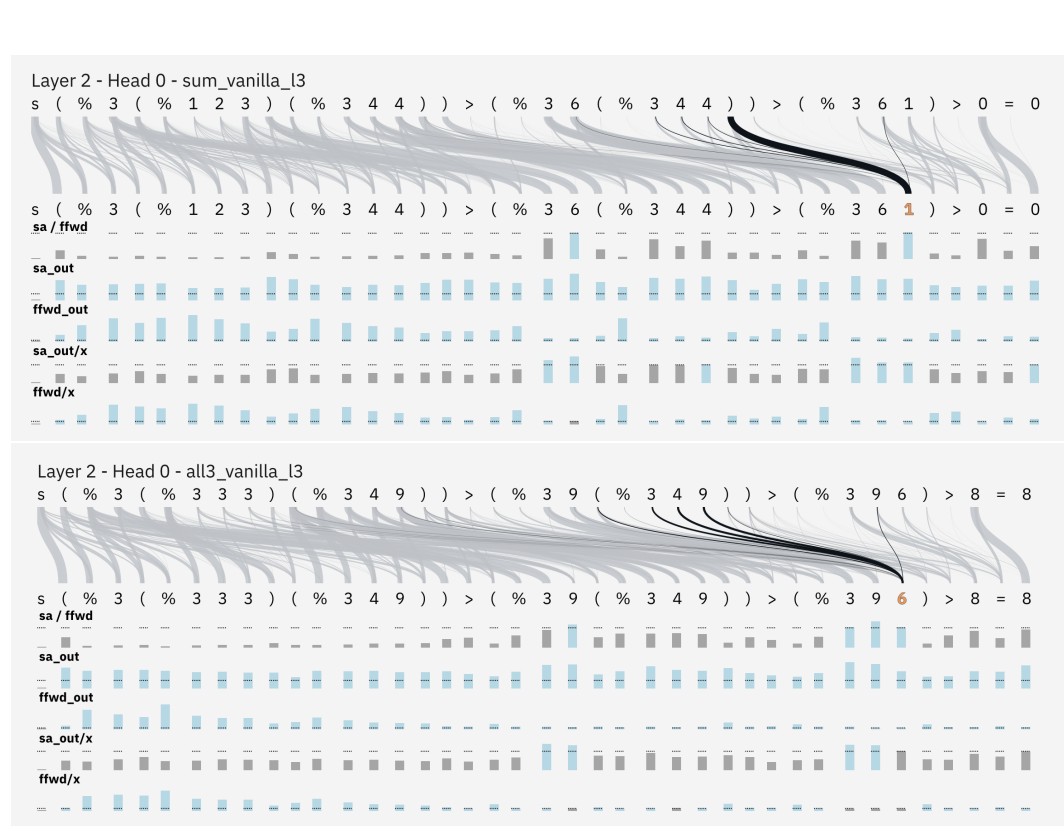

Figure 39: **Attention patterns and layer dynamics in SUM vs MAX+MED+SUM (all3) models.**
Each panel shows (CoT) solution to a SUM modulo 10 problem, where '>' indicates solution
steps. The first row shows the input sequence, with curved lines representing attention weights from
Layer 2 in a 3-layer network. Black lines highlight attention patterns for a specific digit (shown in
orange). Below are shown various layer metrics including the ratio of self-attention to feedforward
norms (sa/ffwd), self-attention output norms (sa_out), feedforward output norms (ffwd_out),
and ratios of layer outputs to inputs (sa_out/x, ffwd/x). **Top:** Model trained only on SUM
operations shows attention primarily focused on parentheses and structural elements. **Bottom:** Model
trained on MAX+MED+SUM (all3) shows attention strongly connecting to digits being combined
in each CoT step, suggesting direct involvement in numerical computation. These distinct patterns
suggest fundamentally different algorithms learned by each model.

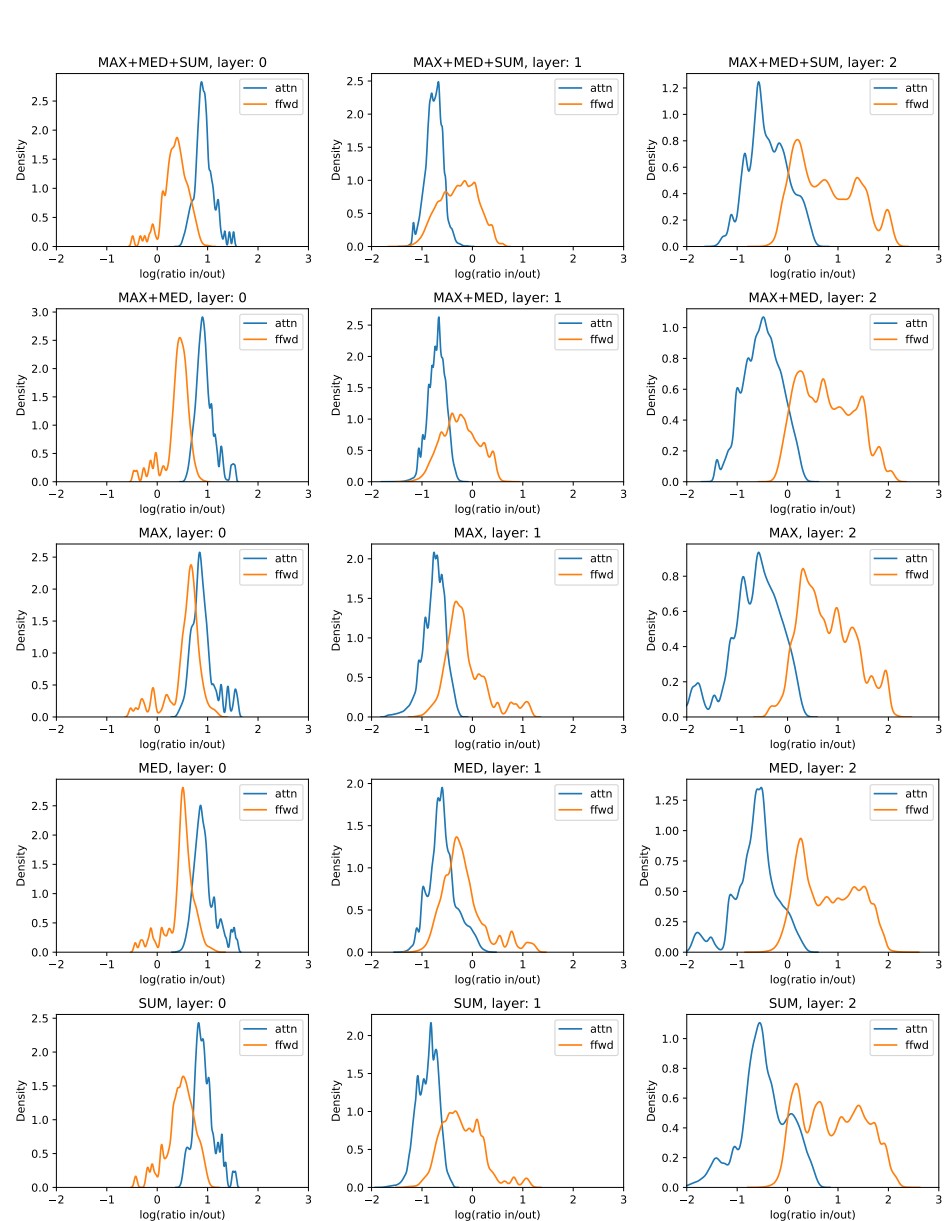

Figure 40: **Attention layer and Feedforward layer ratio histogram.** Each row shows the attention and feedforward layer ratio histogram for models trained on MAX, MED, SUM, MAX+MED,and MAX+MED+SUM. Each column shows the ratio histogram in different attention blocks. The model had 128 embedding and 3 layers and all models reaches 99% accuracy.

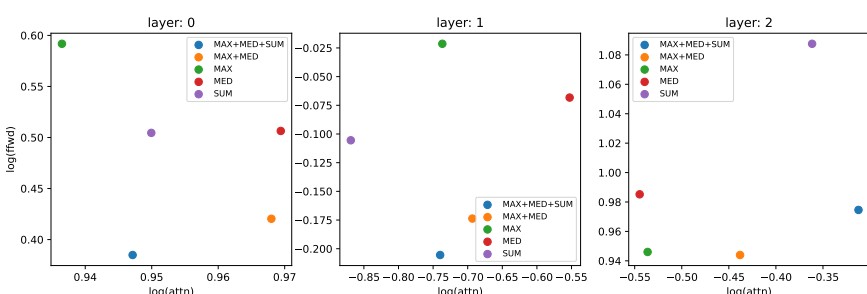

Figure 41: **The mean attention layer ratio vs the feedforward layer ratio.** Each plots show the means in different layers.

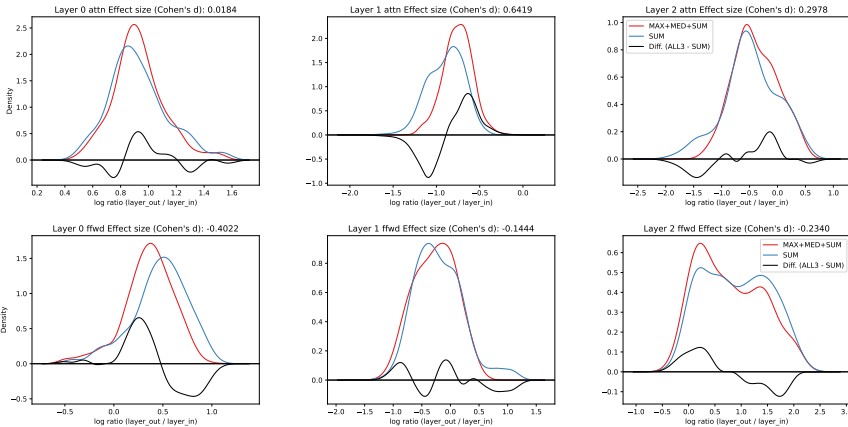

Figure 42: **Comparison of layer output/input ratio distributions** for models trained on all three operations (MAX+MED+SUM) versus sum operation alone (SUM). The title of the figures contain the Effective size showing the difference between the model.

## H COMPLEXITY: NUMBER OF OPERANDS AND NESTING LEVEL.

In previous studies, task complexity has been characterized through various measures, including the number of bits required to memorize the task, which corresponds to the length of the expression Dave et al. (2024), the number of operands and nesting depth Petruzzellis et al. (2024), and the structure of the computational graph Dziri et al. (2024), which captures the number of nestings. Here, we examine how the number of operands and nesting levels affect the learning ability of small language models. Focusing on the `all3` task, which combines `max`, `med`, and `sum` operations, we manipulate complexity by varying the number of operands (`arg = {3, 4, 5}`) and nesting depth (`depth = {3, 4, 5}`). The length of equations, measured by the number of characters, depends on both the number of operands and the nesting level. We find that nesting has a greater impact: increasing the nesting level from 3 to 4 results in longer equations than increasing the number of operands from 3 to 5 at a fixed nesting level. We train the GPT model on the `all3` dataset with all combinations of `arg` (number of operands) and `depth` (nesting levels), finding that the model's performance correlates with the sum of operands and nesting levels (`arg + depth`). Notably, transition points tend to group together for configurations with the same sum (Fig. 43, 44b). While Fig. 44a demonstrates that the model requires more parameters to solve longer equations, it also indicates that `arg + depth` serves as a reliable predictor of the transition point.

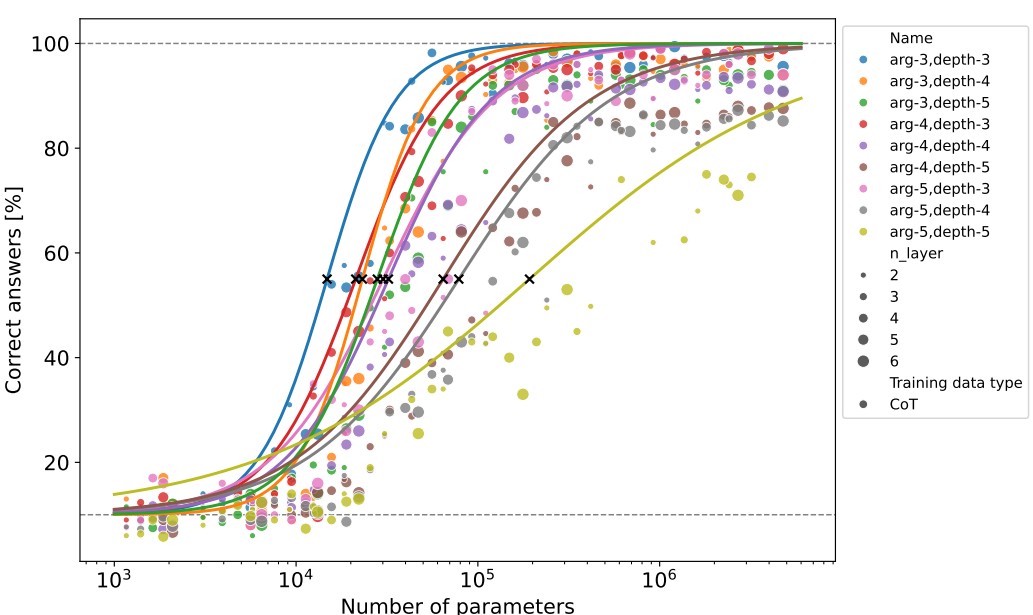

Figure 43: **Learning MAX+MED+SUM operations with varying numbers of operands and nesting levels.** The model requires more parameters as the number of operands and nesting levels increases. Higher nesting levels particularly demand larger model sizes to learn the task. We present the average of five simulations for each configuration and fit a sigmoid function, with the cross marking the middle value (transition point). The transition points reveal an interesting pattern: the sum of the number of operands and nesting levels groups together. For example, the transition points for arg-3,depth-4 (orange) and arg-4,depth-3 (red) are close to each other, as are those for arg-4,depth-5 (brown), and arg-5,depth-4 (grey). Early stopping criteria were applied during training in all simulations.

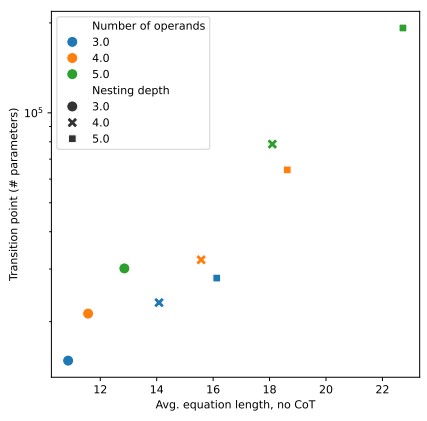

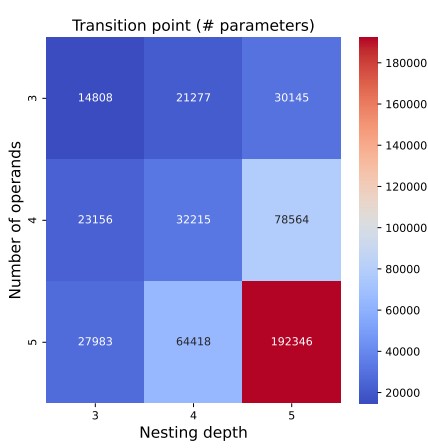

(a) **Average equation length vs. transition point.**

(b) **Transition points.**

Figure 44: **Transition point vs. Equation Lenght and Nesting Depth.** (a) Transition point in function of the average equation length. (b) Heat plot of transition point in function number of operands and nesting depth.

## I CLUSTERING IN THE EMBEDDING SPACE

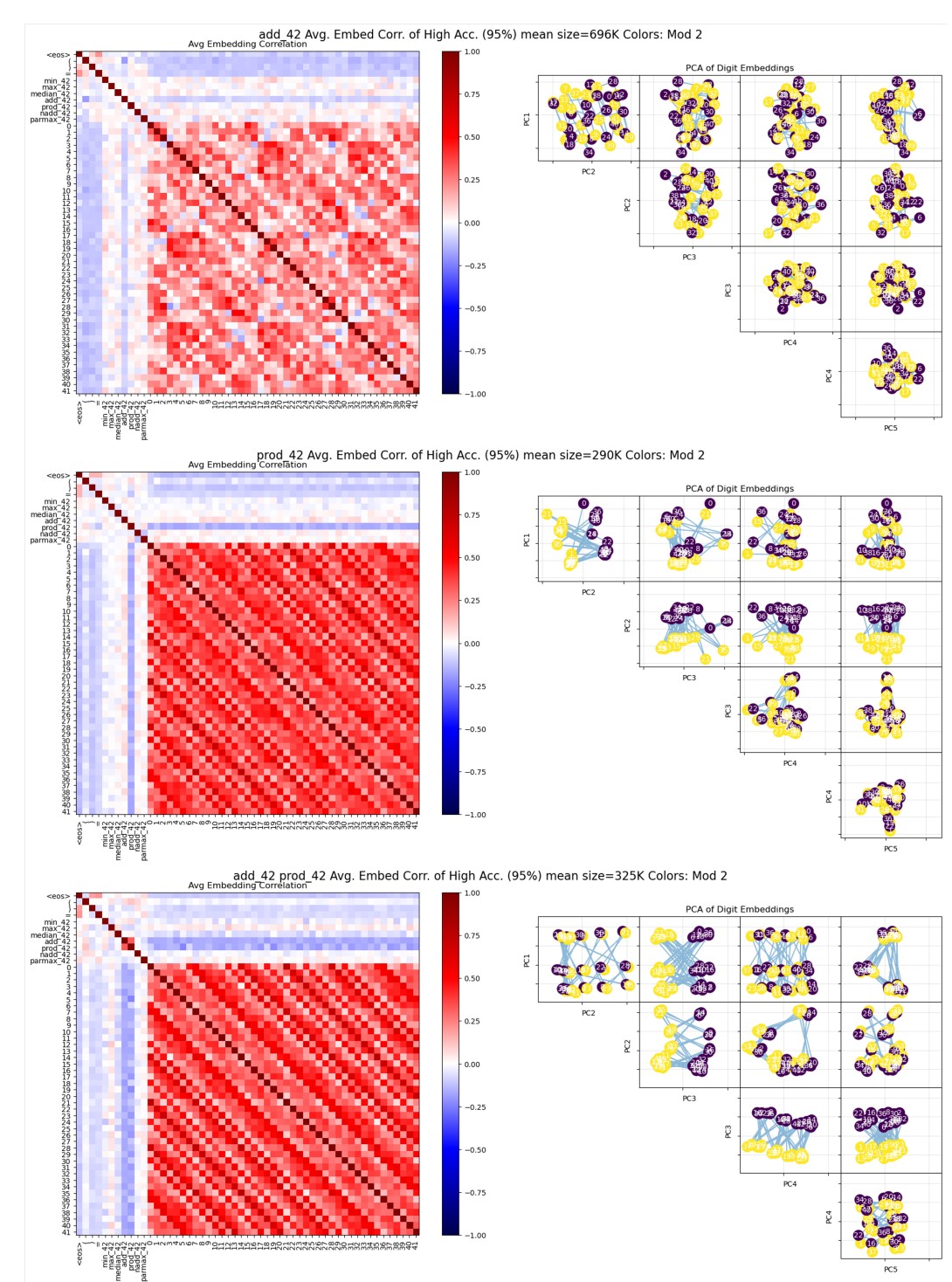

Figure 45: **ADD, PROD and ADD+PRD in MOD 42 PC embeddings**

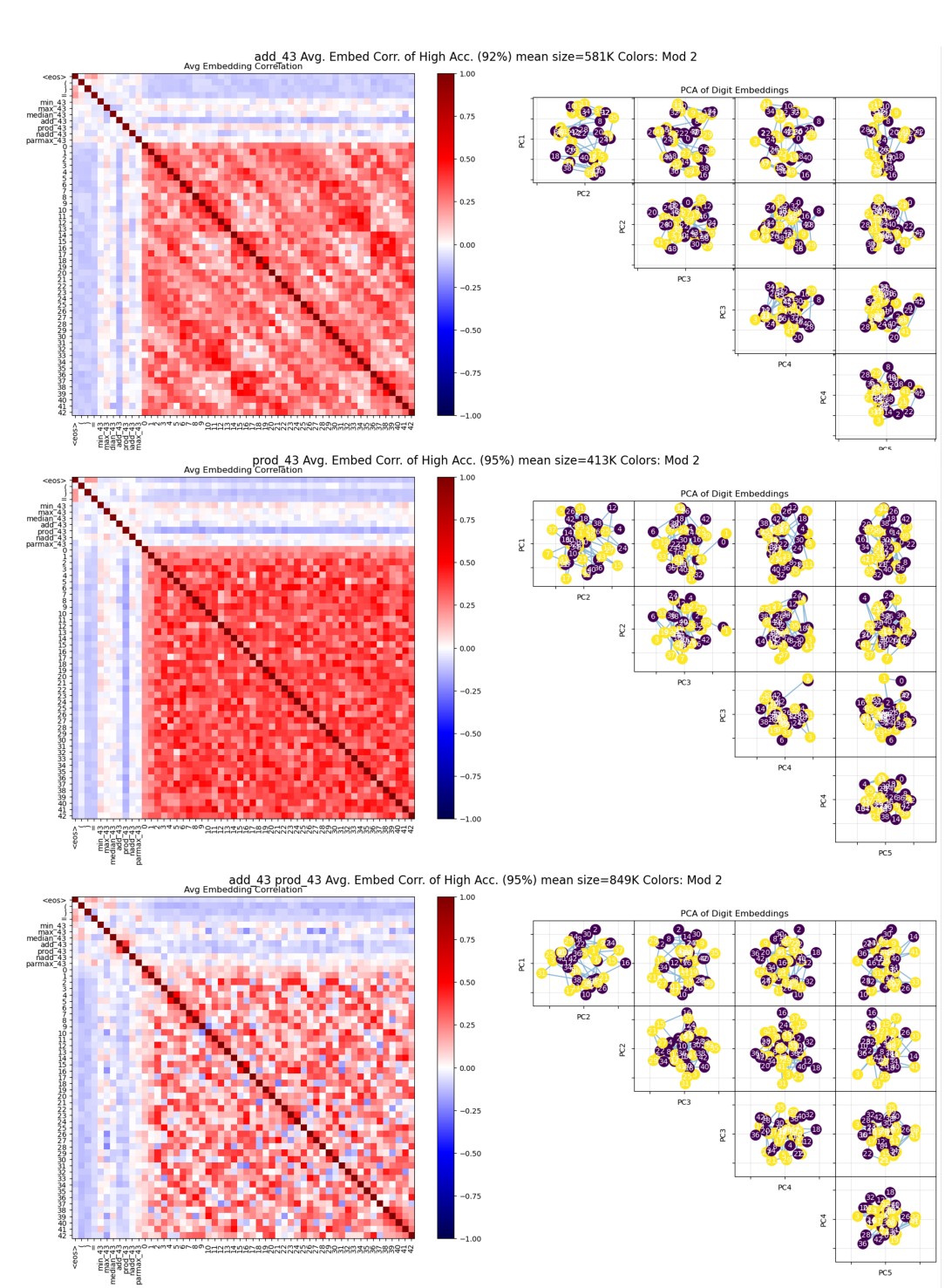

Figure 46: **ADD, PROD and ADD+PRD in MOD 43 PC embeddings**

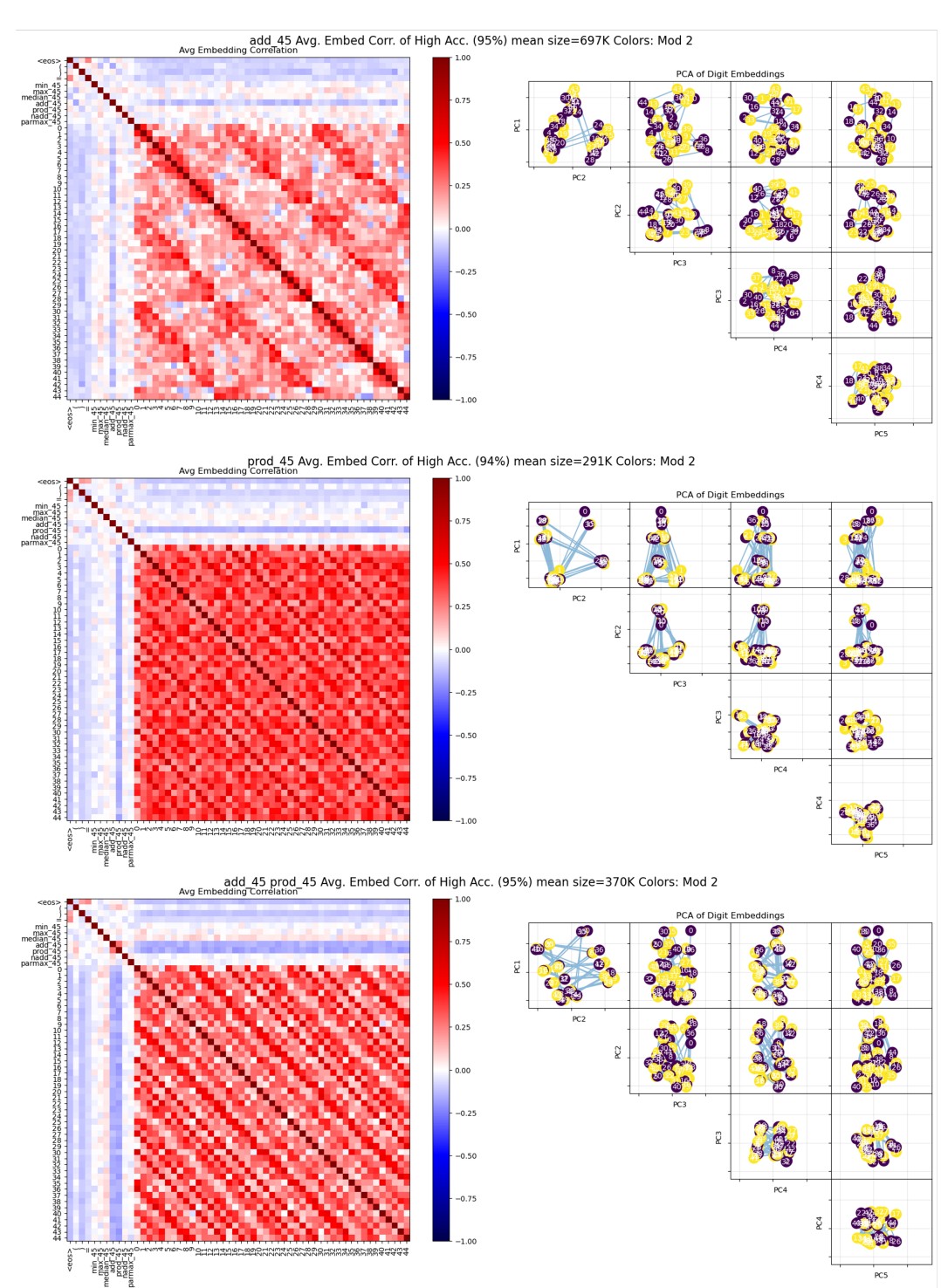

Figure 47: **ADD, PROD and ADD+PRD in MOD 45 PC embeddings**

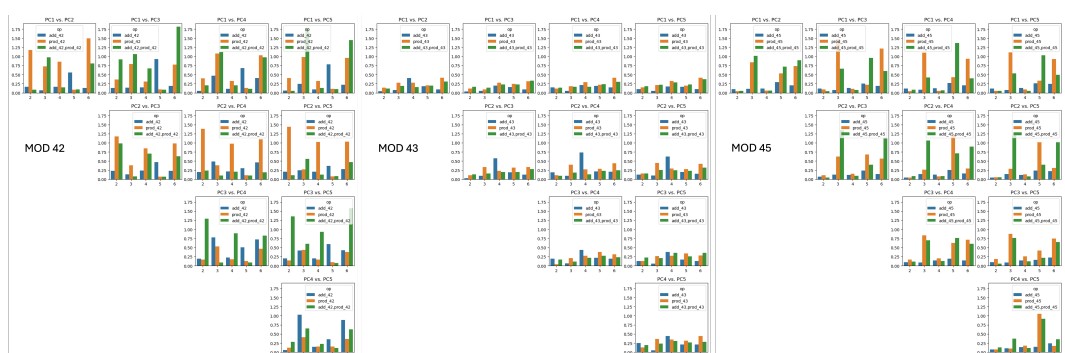

Figure 48: **Clustering in the PC space.** We measure how well modulo-2, 3, 4, 5, and 6 divisor clusters separate in the PC space for the ADD, PROD, and ADD+PROD models on MOD 42, 43, and 45 (for runs with accuracy >80%). Cluster quality is computed from the mean position and standard deviation of each modulo group, and the pairwise distances between them. For MOD 42, strong clustering appears for mod-2, 3, 6, and sometimes 4—precisely the divisors of 42. Strong PROD signals almost always produce strong ADD+PROD signals, whereas strong ADD signals without PROD support do not transfer. For MOD 43, no model shows strong clustering; both ADD and PROD struggle and do not help each other. For MOD 45, strong clustering occurs for mod-3 and mod-5, and occasionally mod-6 (via divisor 3). We also observe that the mixed model only exhibits strong clustering when PROD does as well, indicating that the PROD embedding drives the shared structure and helps ADD.

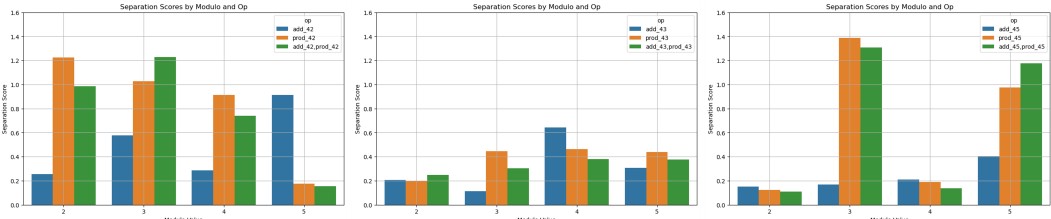

Figure 49: **Clustering in the PC space by averaging over the first 5 PCs.** We measure how well modulo-2, 3, 4, 5, and 6 divisor clusters separate in the PC space for the ADD, PROD, and ADD+PROD models on MOD 42, 43, and 45 (for runs with accuracy >80%). Cluster quality is computed from the mean position and standard deviation of each modulo group, and the pairwise distances between them, averaged over the first five PCs. For MOD 42, strong clustering appears for mod-2, 3, and 4 (could be an artifact of 2) — precisely the divisors of 42. For MOD 43, we find no strong clustering compared to MOD 42 and MOD 45. For MOD 45, strong clustering occurs for mod-3 and mod-5, as expected from the divisors of 45. We also observe that the mixed model exhibits strong clustering only when PROD does as well, indicating that the PROD embedding drives the shared structure and supports ADD.

