# OpenReview forum: "Small Models, Smarter Learning: The Power of Joint Task Training"
_ICLR.cc/2026/Conference — Submitted to ICLR 2026_

### Official Review · Reviewer_5chU · 2025-10-28

**Soundness:** 2
**Presentation:** 3
**Contribution:** 2
**Rating:** 4
**Confidence:** 3

**Summary:**

This work investigates how task difficulty and model size jointly determine a transformer’s ability to learn compositional operations such as modular arithmetic and permutation groups. The authors find that joint training with additional or auxiliary operations can dramatically reduce the parameter requirements, compared to training on isolated tasks. The authors further attribute this improvement to better learned embedding via PCA analysis. They also observed that the training trajectories are quantitatively different.

**Strengths:**

1. The paper presents a very clear setup and is well written.
2. The use of datasets with controllable difficulty provides a strong foundation for conducting rigorous and systematic research.

**Weaknesses:**

My main concern is the limited analysis of model size. The authors constrain the model to a single head and model depth one with recursive reuse, but prior work shows that recursion under a tight parameter budget can severely restrict model capacity even with a similar FLOPs [1], and that the number of heads matters for modular arithmetic datasets [2]. This does not invalidate the authors’ contributions, but a more thorough study of model size, especially head count and depth, would be important.


1. https://arxiv.org/pdf/2507.10524

2. https://arxiv.org/pdf/2502.10390

**Questions:**

I was wondering whether the authors have examined in detail how the mixture of datasets contributes mechanistically to learning. While the paper shows that the embeddings exhibit clear structure, most of the interpretability analysis focuses solely on the embedding level. It would be helpful if the authors could further analyze how the learned features evolve with recurrent depth, whether certain layers or depths specialize in specific tasks. More importantly, I am curious whether any shared computational circuits emerge as the dataset's complexity increases.

Why do the authors present results primarily for non-prime moduli in most plots? Wouldn’t prime moduli represent a more challenging setting, where qualitatively different behaviors might emerge?

---

> ### Author Response · Authors · 2025-11-25
>
> We thank the reviewer for the thoughtful comments. Below, we respond to each concern individually:
>
> ## Weaknesses
>
> **1.1** **Model Size:** Note that our models are already reaching 100\% accuracy in almost all tasks and setting within the model sizes we used. That is why we did not try larger models. We also experimented with different number of layers, architectures, heads etc and got the same results, with the strongest predictor of performance being total number of parameters. Details follow.
>
> **1.2** **Number of heads:**  Thank you for pointing out the number of heads\! All simulations reported in the paper were performed with a single attention head, as we aimed to keep the scratchpad minimal. Following the reviewer’s suggestion, we reran the MAX+MED+ADD and pure-ADD experiments using models with four heads across various model sizes. We observed no differences in performance compared to the single-head models. We will include a new figure in the Appendix comparing the performance of single- and four-head models.
>
> **1.3 Deep transformers:**  Our initial experiments also included standard deep GPT models (nanoGPT; Karpathy, 2022), with results shown in Appendix F. All base (modulo) 26 experiments were conducted with this standard GPT architecture. We adopted the recurrent GPT (Rec-GPT) primarily because it converged faster, required fewer parameters, and produced cleaner structure in the embeddings. In the standard GPT, the embeddings for ADD appear largely unstructured (Fig. 26), whereas Rec-GPT shows early signs of parity separation—with standard GPT, parity was so weak in pure ADD models that we not noticed it and thought parity emerges only under joint training. We found that using more than three recursions generally improves performance, but excessive recursion—such as nine steps—reduces accuracy. All reported simulations used 4 recursions.
>
> We are reorganizing the Appendix to present these results clearly and will reference them in the main text.
>
> ## Questions
>
> **1\. Mechanistical analysis:**  This is very good point. We analyzed the models weights finding:
>
> - **Attention patterns and layer dynamics in ADD vs MAX+MED+ADD:** in the mixed model, the resulting number focussed on the input numbers, whereas in the pure-ADD it focused on the bracket (Fig. 39\)
> - **Activation of attention vs. MLP in ADD vs MAX+MED+ADD:** the mixed model activates the attention weights more than the pure-ADD models, whereas the pure-ADD used more the MLP layer (Fig. 38\)
> - **Circuit**: We did not perform a full circuit analysis but conducted robustness tests. Specifically, we assessed model sensitivity to the removal of attention weights versus MLP weights. We observed no major differences across models; however, models tended to be less sensitive in the deeper layers of standard Transformer setups.
>
> These results are being added to the Appendix for completeness.
>
> **2\. Prime vs. No-prime:** We focus on non-prime modulo in most plots because the ADD operation is a hard task, relatively insensitive to whether the modulus is prime or non-prime, whereas the PROD operation is much more affected by this choice (Fig. 6). Prime modulo present a more challenging setting for PROD, and we have included it on Fig.1 and in the Appendix (Fig.10) to demonstrate the qualitatively different behaviors that emerge.

---

### Official Review · Reviewer_e3ET · 2025-10-29

**Soundness:** 4
**Presentation:** 3
**Contribution:** 3
**Rating:** 6
**Confidence:** 4

**Summary:**

This paper investigates if small models can learn difficult tasks through a synthetic (extension of) ListOps dataset. They discover that joint training with multiple tasks can often lead to efficient learning. They also study embedding patterns to give a 'white-box' explanation for their models.

**Strengths:**

1) The paper studies the effect of joint task learning with multiple experiments, which cover a lot of different aspects of the learning problem (eg. prime v/s non-prime moduli, shuffling of order for ADD, etc.)
2) Interpretability done with embedding vectors was useful. In particular, the restricted embedding hypothesis was a strong evidence for the claims made about the utility of joint training.
3) Experiments on permutation groups present an interesting addition, with a lot of scope for future works.

**Weaknesses:**

1) There is some literature which talks about compositional and/or multi-step mathematical reasoning, the paper was missing references to these [1, 2, 3]. Although the current paper has several important experiments which were missing or not considered in the papers that are mentioned below, it will be useful for the authors to devote some space to discussing these differences in the main text.
2) The last section on 'Discussion and Limitations' doesn't really discuss limitations. For example, these results on synthetic datasets may not transfer immediately to realistic data. The restricted embedding hypothesis, while useful for interp, may not be suitable for practical purposes.
3) Finally, these results on mathematical operations may not necessarily translate to large scale given the widely known problem of arithmetical reasoning in LLMs. This could be alleviated by performing experiments with increasing context length to the maximum limit as allowed by the authors' compute budget.


[1] A. Abedsoltan, H. Zhang, K. Wen, H. Lin, J. Zhang and M. Belkin, “Task Generalization With AutoRegressive Compositional Structure: Can Learning From $D$ Tasks Generalize to $D^T$ Tasks?,” arXiv preprint arXiv:2502.08991, Feb. 2025.

[2] T. Wang and W. Lu, “Learning Multi-Step Reasoning by Solving Arithmetic Tasks,” arXiv preprint arXiv:2306.01707, Jun. 2023.

[3] W. You, S. Yin, X. Zhao, Z. Ji, G. Zhong and J. Bai, “MuMath: Multi-perspective Data Augmentation for Mathematical Reasoning in Large Language Models,” arXiv preprint arXiv:2405.07551, May 2024.

**Questions:**

1) Could you also state that in appendices A.2 contain results for prime moduli (in the first para of Section 3)?
2) Have you tried any experiments where you leverage the ability of the model to learn multiple tasks by then focussing on each of the tasks individually? An example could be to see if further finetuning is necessary for a model to learn just ADD once it has learnt MAX+MED+ADD?
3) Another experiment in the same vein might be to try to finetune models to learn slightly OOD tasks (eg. PROD which is ADD in log space).
4) Finally, if time permits, another interesting experiment to try would be to see what happens if the tasks are presented in a continual learning fashion? Is that as useful as joint learning?

---

> ### Author Response · Authors · 2025-11-25
>
> We thank the reviewer for the positive feedback. And we are excited to implement comments, below we try to address all of the questions.
>
> ## Weaknesses
>
> **1\. Literature review:** Thank you for suggesting the references. We have added them to the new **Related Works** section on page 2 and added a discussion of Multi-Task and curriculum learning.
> While prior work has explored multi-task and curriculum learning, these studies do not examine how model size requirements change under different curricula, nor do they identify which specific task pairings succeed or fail. Although we use a scratch-pad setup similar to grokking studies, our focus is on training small models on combinations of arithmetic tasks. Building on this, we measure how task composition affects the minimum model size required for learning and identify which task combinations help or fail to help models learn hard tasks.
>
> **2\. Discussion and Limitations:** We are extending our discussion section with the following thoughts:
>
> - **Synthetic to real:** While the paper focuses on arithmetic operations, we hypothesize that joint training could be beneficial in more realistic scenarios, such as combining arithmetic with language tasks. Prior research has shown that training arithmetic operations together with text improves model accuracy, and that training on cellular automata patterns can help models perform downstream tasks such as chess move prediction (David S. Yin, Xiaoxin Yin, [https://journals.plos.org/plosone/article?id=10.1371/journal.pone.0310409](https://journals.plos.org/plosone/article?id=10.1371/journal.pone.0310409) , 2024; Lee et al.,[https://arxiv.org/abs/2307.03381](https://arxiv.org/abs/2307.03381) , 2023; Zhang et al.,[https://arxiv.org/abs/2410.02536](https://arxiv.org/abs/2410.02536) ,2025). These examples suggest that models can develop underlying, fundamental representations that can be leveraged to improve performance on downstream tasks.
>
> - **Embedding:** Growing evidence shows that structured, interpretable embeddings help learning. Work on grokking (Power *et al.*, 2022, [https://arxiv.org/abs/2201.02177](https://arxiv.org/abs/2201.02177); Liu *et al.*, 2022, [https://arxiv.org/abs/2205.10343](https://arxiv.org/abs/2205.10343)) reveals that successful generalization coincides with emergence of highly organized internal representations. We suspect that better starting points may lead to better learning. If models begin with structured embeddings or learn them through carefully chosen auxiliary tasks, they may solve problems more efficiently. This suggests that the path to learning matters as much as the final destination: models that develop organized internal representations early in training may require fewer resources to master difficult tasks.
>
> **3\. Context length:**  While our current experiments focus on small models and limited context lengths, we expect that the observed trends, such as the effects of task composition and joint training, will qualitatively hold at larger scales. We tested context lengths between 16 and 512 tokens and found no significant differences above 128 tokens, which is why we chose this range for our experiments. However, this does not rule out the possibility that longer context lengths could influence performance.
>
> ## Questions
>
> **1\. Prime experiments:** We referred A.2 from the main text.
>
> **2\. Individual task evaluation:** While the next-token prediction makes it hard to have a training loss that is only one task, for the validation loss, we evaluate the model on each task separately (Fig. 1, first panel for ADD; Fig. 4 for all tasks), and all tasks reach 100% accuracy individually. Fine-tuning would only be necessary if the model failed to achieve perfect accuracy on a single task.
>
> **3\. OOD finetuning:** Thank you for the suggestion, this is a very interesting direction. Fine-tuning models on OOD tasks (such as learning PROD after training on ADD, given that PROD is ADD in log space) could reveal whether the representations learned transfer to related but distinct operations. If time permits, we design and run this experiment.
>
> **4\. Continual learning:** Thank you for this thoughtful suggestion. Continual learning is closely related to our setting in Sec. 3.1. We implemented a form of continual learning in which we first trained the model on MAX+MED, and after it reached 100% accuracy on these tasks, we slowly introduced the ADD operation. Surprisingly, small models quickly learned the ADD operation, but at the same time, they forgot the MAX+MED tasks. In continual learning, however, the model should retain previously acquired knowledge. This is an interesting and promising direction, and if time permits, we will run additional experiments on this.

---

> > ### Comment · Reviewer_e3ET · 2025-11-26
> >
> > I thank the authors for their comprehensive response. They clearly outline the concerns that were raised by me. I'd like to maintain my positive opinion of this paper and keep my original score.

---

### Official Review · Reviewer_72MW · 2025-10-31

**Soundness:** 2
**Presentation:** 1
**Contribution:** 2
**Rating:** 2
**Confidence:** 2

**Summary:**

This paper studies how small transformer models learn algorithmic tasks, particularly compositional mathematical operations such as those in the ListOps. The authors explore how task difficulty, model size, and training curriculum interact to influence learning efficiency and the emergence of abilities in small models.

**Strengths:**

Quality and Originality: The paper presents a systematic empirical exploration of how joint task training impacts the learning thresholds of small transformer models and attempts to reframe the understanding of scaling laws.

**Weaknesses:**

1. Lack of Novelty. Results are not surprising and authors need to discuss the multi-task learning (i.e., joint training can improve the performance is a common wisdom) and curriculum learning literature. Authors in the introduction discuss scaling laws but they do not provide the exact formulation of laws that contain curriculum learning factors. No precise alternative of KC complexity as well.

2. Small-scale experiments do not support motivations and hurt significance: Authors only conduct experiments on a small scale, not widely used Transformers architecture, and I cannot verify if this can be generalized to larger architectures. Also, based on the text part of Model Architecture, it's hard to reproduce the used model, and the authors should provide more details there (for example, providing a figure or model signature with specific layers will be very helpful to understand your experiments). Larger scale (with more parameters) experiments seem important to verify anything related to "scaling laws".

3. Potentially Trivial Experiments: Not sure if basic arithmetic + permutation group computations (although controllable) are important targets for LLM analysis. When the authors say EASY/HARD tasks, I find it surprising that some of these synthetic tasks are hard for these transformer models and before the Methodology section, I strongly recommend the authors provide preliminary background knowledge to explain the context.

4. Presentation and clarity could be significantly improved. There are too many references to the Appendix without a careful explanation of details about what these experiments are in the main text (For example, line 80 Appendix K) and till line 80 it's hard to understand what is this randomized sum table and its difference with pure SUM.  For permutation groups operations, like OP, it will be much better to provide a concrete example in the main body. Also please fix the presentation of Figure 2 since some texts now overlap.

**Questions:**

1. How well do these findings transfer to natural language or real-world multi-task learning scenarios? Demonstrating such transfer would strengthen the broader relevance of the conclusions.
2. Can the authors formalize or quantify how task combinations interact to better predict which combinations yield synergy versus interference?
3. Can the authors situate their findings within the broader context of several key work of scaling law theory and at least think about how to incorporate the missing parts like curriculums?

---

> ### Author Response · Authors · 2025-11-25
> **Part 1**
>
> Thank you for your comments. Let us clarify the novelty. We hope the reviewer will share our excitement about these results. It is true that the benefits of joint training have been widely observed empirically. However, what we show is the **"Joint training paradox":** the choice of training curriculum can be a vastly more important factor than previously quantified.
>
> ## Weaknesses
>
> **1.1. Lack of Novelty:** To our knowledge, the dramatic effect and the paradox in joint training has not been quantified before:
>
> * **The Paradox:** Mixing certain **"easy" tasks** (like MAX, MIN, or PROD) with otherwise **"hard" tasks** (like modular addition or permutation group products) allows the model to learn the hard task at a significantly smaller parameter size.
> * **Magnitude:** We quantified this effect, showing that joint training **reduces the minimum parameter requirements** for the hard tasks by a factor of **2x to 7x**.
> * **Failure of joint training:** Crucially, we find that mixing **two hard tasks** together does *not* seem to produce this benefit. Counterexamples include:
>   * ADD+NADD
>   * mod prime PROD+ADD
>   * Shuffle SUM \+MAX \+MED
> * **Mechanism:** we narrow down the mechanisms
>   * **Structured embedding:** usually associated with this effect
>   * **subtasks in the composite task:** we construct examples, e.g. permutation group
>
> **1.2. discuss the multi-task and curriculum learning literature:** Good suggestion, we are adding more related work. Specifically:
>
> * **MTL eavesdropping:** task interactions can allow one task to “eavesdrop” on learned features of other tasks (Ruder, 2017, “An Overview of Multi-Task Learning …”)
> * **Strategic Data Ordering (Kim et al., 2024):** This work proposes curriculum learning strategies that order LLM training data based on complexity metrics, demonstrating that curriculum learning can slightly improve performance compared to random data shuffling on models like Mistral-7B and Gemma-7B.
> * **What Makes a Good Curriculum? (Jia et al., 2025):** finds that no curriculum strategy dominates universally—the relative effectiveness of forward (easy to hard) versus reverse (hard to easy) curriculum learning depends jointly on model capability and task complexity, with stronger models benefiting from forward curriculum on simpler tasks while weaker models or harder tasks often favor reverse ordering.
> * **Beyond Neural Scaling Laws via Data Pruning (Sorscher et al., 2022):** shows that with high-quality data pruning metrics that rank training examples, it's possible to break beyond power law scaling and potentially achieve exponential scaling instead. This suggests that data quality and ordering matter significantly for scaling behavior.
>
> Many of these results are complementary to ours. **None of them performs a systematic analysis at different model sizes**, pretrained from scratch. Sorscher 2022 and Kim 2024 focus on the choice of data in a continuous pool and don’t examine details of synergy between “tasks”. Jia 2025 varies the problem difficulty, but again does not vary the model sizes and does not train from scratch.
>
> **1.3. Scaling laws for curriculum:** To our knowledge, no scaling law taking curriculum into account has not been formalized. Sorscher 2022 shows that we can get exponential scaling instead of power laws with proper data pruning. But synergy and interaction between different tasks can occur on multiple features (e.g. numbers in our tasks, or other concepts in language tasks) and it is not clear whether finding a unified theory is even possible. Our focus is not the scaling laws, and we only wish to highlight that the shift in the emergence of abilities transition may affect scaling laws, similar to Sorscher 2022\.
>
> **1.4. Alternative to Kolmogorov Complexity:** As in our intro, we are not using KC. Our point is that brute-force vs smarter algorithmic solutions have different description lengths and learning them requires different model sizes. A proxy for the description length can be the zipped size of the code written in a very low-level programming language (e.g. only using bitwise ops in Assembly).
>
> **2.1. Small scale experiments:** Our models are small because the transitions happened at these sizes. In our tests, any model larger than these sizes was able to learn the task. So there was no need to go any bigger.
>
> **2.2.  Not widely used transformer arch.:** Actually, we do have many experiments with the usual deep transformer model (All Mod 26 experiments in the appendix) and we observed the same effect. While not as popular, recurrent GPT models are used in Google PaLM models, and introduced in GPT-J. They are a special case of Universal Transformers.
> The reasons we switched and mainly used rec GPT are detailed in the general comment above and included cleaner embedding structures, faster convergence and having a single layer which made analyzing the weights easier.

---

> > ### Author Response · Authors · 2025-11-25
> > **Part 2**
> >
> > **2.3. Hard to reproduce arch:** We are adding a sketch of the model. We also forgot to share our code. We have now added that in the supplementary. Also, adding the hyperparameters to the appendix. Each experiment takes just minutes to run and they are fun to watch. We highly recommend playing with the code\!
> >
> > **2.4 Larger models needed for scaling laws:** The scaling laws are not our focus, but we have scaled in other ways. In Fig. 6: **Scaling of transition point vs input range (modulo)** we scale the modulo and observe separate trends for prime and non-prime modulos. But for a fixed modulo there are no more experiments to do, as the models already pass the learning threshold.
> >
> > **3\. Potentially Trivial Experiments… recommend background:** Sure, we are adding more background on this. Surprisingly, these arithmetic tasks have been long known to be challenging for LLM. They have been used frequently for LLM analysis, from the original emergent abilities paper ([Wei 2022](https://arxiv.org/abs/2206.07682) ), to grokking ([Power 2022](https://arxiv.org/abs/2201.02177)). Earlier works had shown that LLM are bad at arithmetic ([Maxwell et al. 2021, “Show your work…”](https://arxiv.org/pdf/2112.00114)). Teaching arithmetic ([Lee 2023](https://arxiv.org/pdf/2307.03381)) and understanding how models encode numbers ([Zhong Neurips 2023 the clock and the Pizza](https://proceedings.neurips.cc/paper_files/paper/2023/file/56cbfbf49937a0873d451343ddc8c57d-Paper-Conference.pdf)).
> >
> > Additionally, we want to point out that ListOps does not need to be arithmetic. Similar to our permutation example, it can be generalized to arbitrary functions on sets (including procedural or formal language).
> >
> > **4\. Presentation and clarity:** Thank you, we are improving it. We are adding new figures to explain:
> >
> > 1. Shuffled SUM process
> > 2. Sketch of Rec GPT
> > 3. Sketch of permutation OP\_top and OP\_bottom
> >
> > We will also give examples in the main body.
> >
> > ## Questions
> >
> > 1. **Transfer to natural language or real-world multi-task scenarios?**
> >    A recent paper, “Intelligence at the Edge of Chaos” (Zhang ICLR 2025 https://arxiv.org/pdf/2410.02536) provides fascinating clues about this: They trained models on cellular automata data and found that the models became much smarter in Chess. We are adding this to the paper. We believe that there may be exciting effects in real-world tasks, similar to our joint training observations. The crucial next step would be to find the task complexity and synergy measures which would result in this effect. In the current paper we only wanted to document quantitatively that joint training can have a dramatic effect on model size and understand some of the mechanisms behind it.
> > 2. **Formalize … to predict which combinations yield synergy versus interference?**
> >    Great question\! It seems tasks which help to “divide and conquer” a hard task could be beneficial (similar to feature eavesdropping in MTL). Ex: ADD , A+B=C, has a uniform distribution for C, making it hard because there is no pattern that stands out. PROD (non-prime) or MAX can help ADD probably by *breaking this symmetry*, as they have a peaky output distribution (e.g. in MAX the output contains larger numbers more). NADD has a similar uniform distribution as ADD ( `(A-B)%n= (A+(n-B))%n` ) so it doesn’t help ADD.
> >    Our goal here was to quantify this effect with concrete examples. Attempting to formalize this effect is a great future research direction.
> > 3. **Situate findings within the broader context of key works of scaling law theory and think about how to incorporate the missing parts like curriculums?**
> >    This is a really good point. Attempting to include curriculum in scaling laws may be complicated, as task interaction can occur along different directions (see our answer to 1.3 in “weaknesses”). There is no formal result on curriculum or multi-task learning. The key observations have been Sorscher 2022 where pruning can *change* the scaling law from power law to exponential.
> >    Our work explores an orthogonal direction to the past work: How does task synergy and overlap modify the learning transition. One way to include the curriculum is to explore compositionality: If it is true that composite tasks become easier when models are trained on their component tasks then one way to extend scaling laws could be redoing measure the scaling alone and with the composite tasks. Since our focus was not the scaling laws we didn’t explore that. But we can measure it and get back to you.

---

### Official Review · Reviewer_S5KR · 2025-11-01

**Soundness:** 3
**Presentation:** 2
**Contribution:** 2
**Rating:** 4
**Confidence:** 4

**Summary:**

This paper investigates how joint task training can reduce the parameter requirements for learning arithmetic tasks. The authors also provide a preliminary mechanistic explanation suggesting that joint training leads to more structured embeddings.

**Strengths:**

The paper presents several interesting phenomena

- The embeddings of jointly trained models separate even and odd numbers on ADD and PROD tasks.
- The experiment on shuffled SUM suggests that the benefit of joint training emerges only when the easy and hard tasks share underlying numerical properties (Figure 3).
- The transfer learning experiment demonstrates that pretraining on simpler tasks and then transferring to harder ones is an effective curriculum. (Figure 5)

**Weaknesses:**

- The novelty and significance of the work may be limited.
  - The general observation that joint training or curriculum learning benefits language models, including for arithmetic tasks, has been reported previously. The most related prior work I know is [1]; it would be helpful for the authors to clarify how their approach and findings differ from that work.
  - While the finding that joint training leads to more structured embeddings is interesting, the paper does not analyze how these embeddings form or how they influence the parameter requirements of learning the task.
- It would be helpful if the authors could more explicitly summarize the main takeaways, epspecially how the findings on synthetic tasks might transfer to more realistic scenarios.
- The transfer learning experiment in Section 3.1 does not convincingly support the “embedding-restriction” hypothesis to me, as the results do not directly demonstrate that the effective search space is reduced.

- The paper would benefit from including at least a minimal related-work section in the main text for better contextualization.
- Minor comments (do not affect the score):
  - l161 "sizewe" -> "size we".
  - The interchangeable use of *SUM* and *ADD* could be confusing; I recommend using one consistently.
  - Figure 2a: the two subplots overlap.
  - Figure 2 caption (line 266): “(c)” should likely be “(b)”.
  - l348 "The also show".

**Questions:**

- Why did the authors choose to use a recurrent version of the Transformer instead of a standard multi-block architecture? Would the results remain consistent under a conventional architecture? It would be useful to clarify this choice more explicitly.

- How did the authors determine the performance for a fixed number of parameters? As discussed in Section 4, even after convergence, grokking may occur for arithmetic tasks.
- The jointly trained embeddings reportedly separate even and odd numbers. Do the authors have an explanation for why such parity patterns emerge in the ADD and PROD tasks, given that parity does not appear to play a direct role in either task?

---

> ### Author Response · Authors · 2025-11-25
> **Part 1**
>
> We would like to thank the reviewer for taking the time to read our paper carefully and provide critical feedback. We address these points below, one by one:
>
> ## Weaknesses
>
> **1.1 Novelty and significance:** Thank you for the question. We would also like to ask the reviewer to share the recommended paper, as it was not attached to the review.  For clarity, we summarize our contribution below:
>
> Earlier studies have focused on curriculum-like strategies for arithmetic, including difficulty ordering and data-format changes such as reversed outputs or chain-of-thought supervision, showing that such approaches allow models to reach higher accuracy with substantially less data and fewer parameters.
>
> Our work addresses the **joint-training paradox** from the perspective of task synergy: which combinations of tasks enable smaller models to learn, and why? We identify that:
>
> * **Easy \+ Hard:** pairing specific “easy’’ tasks (such as MAX, MIN, or non-prime PROD) with “hard’’ tasks (such as modular ADD or permutation products) **allows models to learn the harder task with models that are 2–7× smaller.**
> * **Hard \+ Hard:** Crucially, this effect is highly selective: pairing two hard tasks—such as ADD+NADD, prime-modulo multiplication with ADD, or shuffled ADD with MAX and MED—**offers no benefit**.
> * This selectivity shows that joint training succeeds only when the tasks bear meaningful structural relationships, rather than simply increasing training diversity. We identify two mechanisms behind this phenomenon:
>   * **Successful pairings produce structured numerical representations** in the principal components of the embedding space.
>   * **The easier tasks often contain natural subtasks of the harder ones**. This principle extends beyond arithmetic to permutation groups, where learning block operations helps models acquire full-group products.
>
> **1.2** **Embedding evolution**: Very good question; we believe we already have the answer. Fig. 4 illustrates the evolution of the embedding for MAX+MED+ADD and for pure ADD, along with the model accuracy on the respective tasks. We find that:
>
> * **The embedding undergoes a transition** from a noisy (random) state to an organized structure (arc or grid-like), reflecting **number ordering and parity separation**.
> * **As this structure forms, the accuracy also increases**, indicating that the emergence of a structured embedding **correlates with the model’s emergent abilities**.
> * **No structure \==\> Higher param count:** Furthermore, we also observe that when the model fails to develop a well-organized embedding, such as in Shuffled-ADD, prime-PROD, or ADD, it requires a substantially larger number of parameters to learn.
>
> Our empirical results demonstrate that, under joint training, these hard tasks benefit from the organized embeddings induced by the easy tasks, thereby reducing the required model size.
>
> **2.1. Takeaways:** Good suggestion. We have listed the main takeaways in the “General comment” above. The key points are:
>
> 1. **Paradox:** Easy subtask+hard task beneficial; two hard tasks no helping one another,
> 2. **Embedding structure:** coinciding with reduced model size,
> 3. **Shuffled SUM** not benefitting from MAX, MED, supporting role of **number properties**
> 4. **Scaling with modulo:** showing robust effect across scales and ops
> 5. **Generalization:** beyond arithmetic using permutations
>
> As summarized above, our work empirically demonstrates how task synergy can enable more parameter-efficient learning of hard tasks. We show clear positive examples—such as ADD+MAX+MED, MAX+prime-PROD, and ADD+non-prime-PROD—where joint training helps smaller models learn tasks that they cannot learn in isolation. We also provide counter-examples, such as ADD+NADD and prime-PROD+ADD, where joint learning does not provide benefits. Furthermore, in cases where joint training is beneficial, we observe that the model develops a more structured embedding space. We also find that composable tasks can support each other: a complex task can be facilitated by its constituent subtasks, as demonstrated in our permutation experiments.
>
> **2.2. Transferability:** While the paper focuses on arithmetic operations, we hypothesize that joint training could be beneficial in more realistic scenarios, such as **combining arithmetic with language tasks**, as well as **using rule-based data for pretraining.** Some relevant works are:
>
> * **Training arithmetic operations together with text** improves model accuracy, ([Lee 2023](https://arxiv.org/abs/2307.03381); [Yin & Yin 2024](https://journals.plos.org/plosone/article?id=10.1371/journal.pone.0310409))
> * **Training on cellular automata patterns** can help models perform downstream tasks such as **chess move prediction** ([Zhang 2025](https://arxiv.org/abs/2410.02536)).
>
> These examples suggest that models can develop underlying, fundamental representations that can be leveraged to improve performance on downstream tasks.

---

> > ### Author Response · Authors · 2025-11-25
> > **Part 2**
> >
> > **3\.** **Embedding restriction**: We actually do have results on evolution of embedding: Fig. 4 compares the embedding trajectories for MAX+MED+ADD versus pure-ADD, alongside the corresponding task accuracies. We observe that:
> >
> > * The **pure-ADD model**, without the inductive biases introduced by MAX and MED, **converges much more slowly** toward an embedding configuration from which learning can begin.
> > * This suggests that **joint training can guide the model into a more favorable region of the embedding space**, effectively reducing the search burden even if the restriction of the search space is not shown directly.
> >
> > We are also adding new results in the revision where we directly track the embedding evolution in settings where joint training is beneficial and in settings where it is not.
> >
> > **4\. Related work**: We agree with the reviewer. We are adding a discussion of these papers to the new **Related Works** section on page 2 (see updated paper). We have also added more to the discussion section.
> > While prior work has explored multi-task and curriculum learning, these studies do not examine how model size requirements change under different curricula, nor do they identify which specific task pairings succeed or fail. Although we use a scratch-pad setup similar to grokking studies, our focus is on training small models on combinations of arithmetic tasks. Building on this, we measure how task composition affects the minimum model size required for learning and identify which task combinations help or fail to help models learn hard tasks.
> >
> > **5\. Minor:** Thank you for pointing out these mistakes. We will address them.
> >
> > ## Questions
> >
> > **1\. Model architecture:** Thank you for the question. Yes, our findings remain consistent when using a standard multi-block Transformer architecture. In fact, our initial experiments were conducted with regular GPT models (nanoGPT; Karpathy, 2022), and the corresponding results are provided in Appendix F. All modular-26 experiments also uses the standard GPT architecture.
> >
> > We adopted the recurrent GPT (Rec-GPT) primarily because it converged faster, required fewer parameters, and produced cleaner structure in the learned embeddings. In the standard GPT, the embeddings for ADD (SUM) appear largely unstructured (Fig. 26), whereas Rec-GPT shows early signs of parity separation—in standard GPT we noticed parity only under joint training. As expected, the parity signal becomes much stronger in the joint-training setting.
> >
> > Although less widely used, Rec-GPT models appear in Google’s PaLM family and were introduced in GPT-J; they can be viewed as a special case of Universal Transformers. We will include a brief sketch of this architectural choice in the Appendix E.
> >
> > **2\. Grokking:** We report accuracy on a held-out test set. Unlike classical grokking behaviour—in which test accuracy stays low for a long period before a sudden jump—our models begin to generalize early in training (Fig. 4), even though the test set contains triplets never observed during training. This difference likely stems from the data distribution: in our setting, most number pairs appear intermittently throughout the training stream, whereas grokking studies typically use inputs consisting of a *single* pair of numbers with very limited coverage. Consequently, we do not observe grokking dynamics in our models; test accuracy rises too early for a grokking-like phase transition to occur. We also trained small-parameter models (where the model did not learn up to 20k steps) for up to 50k steps and observed no improvement in performance.
> >
> > **3\. Parity:** While we don’t have a proof, we strongly suspect the algorithm *does use* these features. For instance, the models could be using a divide and conquer strategy (odd+odd=even, even+odd=odd, even+even \= even). For the PROD task, the embeddings reveal higher-order modular structure (Fig 10). For example, when examining mod 12, numbers cluster according to mod 2, 3, 4, and 6.

---

### Author Response · Authors · 2025-11-25
**General Comment**

We thank all reviewers for their insightful comments. We are pleased to see the reviewers find our observations interesting and don't find major flaws with the analysis. We are addressing the questions and criticisms individually. Let us also summarize the answer to a few common questions in this comment.
**Edits in the paper are written in blue.**

## Key Takeaways and novelty

1. **Joint training Paradox:**  Mixing certain **"easy" tasks** (like MAX, MIN, or PROD) with otherwise **"hard" tasks** (like modular addition or permutation group products) allows the model to learn the hard task at a significantly smaller parameter size. More details below.
2. **Novelty:** To our knowledge, we are the first to quantify the effect of joint training in terms of
   1. **Magnitude:** reporting 2-7x reduction in model size when it's beneficial.
   2. **Failure cases:** We find explicit cases where joint training hurts learning
   3. **Beyond arithmetic:** We show the effect also in permutation matrices and believe it extends to other systems.
3. **Mechanisms:** We observe two mechanism behind the effect:
   1. **Structured Embedding:** beneficial joint training leads to both smaller models fewer training steps. Fig 4 shows emergence of good embedding coincides with learning the task.
   2. **Attention vs MLP utilization:** We have limited results from SUM (ADD) vs MAX+MED+SUM, suggesting SUM alone model relies slightly more on the MLP module than the attention compared to the joint model (Appendix G, Fig 42).
4. **Shuffled-SUM:**
   1. Shuffled SUM alone is as hard as SUM alone
   2. **MAX+MED+Shuff-SUM fails to learn:** surprisingly, mixing Shuffled SUM with MAX and MED leads to the models **never learning to do Shuffled SUM.** This is in sharp contrast with mixing SUM with MAX+MED, which leads to small models perfectly mastering SUM.
   3. Strongly suggests the number features learned from MAX+MED actually hurt learning Shuffled SUM, which does not use any number properties.
5. **Scaling with modulo:** The effect is robust across different scales for the modulo


## Paradox of joint training

Joint training does not always help. We have found multiple examples where it helps and some very interesting cases where it hurts:

1. **Positive cases (easy+Hard tasks):** Mixing a subtask or a related, easier task makes the hard task easier
   4. **Easy:** MAX, MIN, MED or PROD (non-prime mod), \+ **Hard:** ADD, NADD
   5. MAX+PROD (prime mod)
   6. **Block-diagonal 6x6 Permutations: Easy:** Single 3x3 block permutation \+ **Hard:** 6x6 two blocks permutations
5. **Negative cases (Hard+Hard tasks):** Mixing **two hard tasks** together does *not* seem to produce this benefit. Counterexamples include:
   1. ADD+NADD
   2. mod prime PROD+ADD
   3. Shuffle SUM \+MAX \+MED

## Transferability

While the paper focuses on arithmetic operations, we hypothesize that joint training could be beneficial in more realistic scenarios, such as **combining arithmetic with language tasks**, as well as **using rule-based data for pretraining.** Some relevant works are:

* **Training arithmetic operations together with text** improves model accuracy, ([Lee et al 2023](https://arxiv.org/abs/2307.03381); [Yin & Yin 2024](https://journals.plos.org/plosone/article?id=10.1371/journal.pone.0310409))
* **Training on cellular automata patterns** can help models perform downstream tasks such as **chess move prediction** ([Zhang et al. 2025](https://arxiv.org/abs/2410.02536)).

## Architecture

The recurrent GPT is not essential for the effect, neither is the single head. We did have many experiments with the **usual deep transformer model** (All Mod 26 experiments in the appendix) and we observed the same effect. While not as popular, **recurrent GPT models are used in Google PaLM models, and introduced in GPT-J**. They are a special case of Universal Transformers.

The reasons we switched and mainly used rec GPT were:

1. Its single layer has fewer weight matrices, easier for interpretability.
2. Its embedding layer showed stronger, cleaner structures
3. It converges more consistently, in fewer iterations, and smaller models learn the tasks.
4. Even on pure ADD (SUM) it partially learns parity, challenging our initial observation from deep transformer model experiments that SUM embedding was completely random.

We also did many **experiments with multiple heads** and observed the same effect without significant change to the transition point. We are adding those results to the appendix.

---

### Author Response · Authors · 2025-11-30
**Code**

We have now shared the code in the Supplementary Material.

---

### Author Response · Authors · 2025-12-03
**Summary of all revisions**

We thank all the reviewers for their helpful feedback. We would like to take a moment to update the new AC on the changes and improvements, as well as response to major points during the rebuttal.

**Clarification of Novelty and Takeaways:** Our paper focuses on quantifying the joint training paradox – we show exactly that mixing certain easy tasks (MAX, PROD) with hard tasks (modular ADD, permutation group products) allows the model to learn the hard task at a significantly smaller parameter size (2-7x reduction in model size). We also show that the emergence of structured embedding coincides with learning the task. We demonstrate the joint training paradox beyond arithmetic using permutation groups.

**Missing related work:** We have added a section to the paper discussing joint-training of arithmetic operations (Abedsoltan et al., 2025; Lee et al., 2023; Wang & Lu, 2023), curriculum learning (Yin et al., 2024; Kim et al., 2024; Jia et al., 2024), and the formation of highly structured internal embeddings in models solving modular arithmetic (Power et al., 2022; Liu et al., 2022; Nanda et al., 2023). To our knowledge, none of the previous works perform a systematic analysis of the effect of joint training at different model sizes, pretrained from scratch, and show examples of successes and failures of joint training.

**Transferability:** We have added a section to the discussion about combining arithmetic with language tasks (Lee et al., 2023; Yin & Yin, 2024), as well as using rule-based data for pretraining (Zhang et al., 2025).

**Embedding restriction:** Fig. 4 compares the embedding trajectories for MAX+MED+ADD versus pure-ADD, alongside the corresponding task accuracies, showing that the pure-ADD model converges slowly toward an embedding configuration from which learning can begin. In contrast, joint training can guide the model into a more favorable region of the embedding space, effectively reducing the search burden even if the restriction of the search space is not shown directly.

**Model architecture:** We chose the recurrent transformer model (faster convergence, fewer parameters, more structured final embedding). We have added a diagram of the model to the Appendix. **Depth and** **Head number:** We ran more simulations showing that the results are not impacted by either the number of heads or by the number of transformer blocks in the traditional sequential transformer setup.

**Parity separation in the embedding:** We show that in the embedding, the model groups the numbers based on the divisors of the modulo: prime mod PROD yields non-structured embeddings, even mod PROD yields grouping by parity and higher divisors, and odd mod PROD yields similar grouping without parity separation.

**Scaling laws for curriculum:** We add clarification in our answers that, based on our knowledge, no scaling law taking curriculum into account has been formalized, and our focus is not on scaling laws. We only wish to highlight that the shift in the emergence of abilities may affect scaling laws.

**Mechanistic analysis:** We observe two mechanisms behind this effect. First, the emergence of good embeddings coincides with learning the task. Second, we observe patterns in attention vs. MLP layer utilization; for example, pure ADD utilizes the MLP layer more heavily, while the MAX+MED+ADD model relies more on the attention layer.

---

### Meta-Review · Area_Chair_GfeV · 2026-01-07

**Summary:**

This paper investigates how joint training on compositional tasks (e.g., ADD, PROD, MAX on ListOps) reduces minimum model size requirements, reporting 2-7x parameter reductions when training hard tasks jointly with easier ones. The authors analyze embedding structures via PCA, showing joint training produces more structured representations than single-task training, and extend findings to permutation groups. Most reviewers were not recommending acceptance, with consensus concerns about limited novelty relative to multi-task learning literature, too small experimental scale for claims about LLMs and scaling laws, and lack of mechanistic understanding beyond embedding analysis.

**Reviewer Concerns:**

The rebuttal tried to address some key concerns, but they seem not fully resolved yet. (1) Joint training benefits are well-known, and the work needs to be more adequately positioned. (2) Experiments are only on tiny models. (3) Extensive experiments with various architectural choices are missing. (4) Curriculum learning baseline is missing. (5) Limited generalizability analysis, i.e., unclear how findings transfer to realistic scenarios or larger scales.

**Reviewer Scores:**

Reviewers concerns were partially addressed, but the key concerns are not fully addressed.

---

### Decision · Program_Chairs · 2026-01-26

Reject